# Trichiasis with and without tarsal conjunctival scarring: A multi-country observational study

Esmael Habtamu[1,2,3]*, Caleb Mpyet[4,5], Francis Mugume[6], Gladys Atto[7], Fikre Seife[8], Gilbert Baayenda[9], Sharone Backers[10], Ana Bakhtiari[11], Zerihun Tadesse[12], Scott D. Nash[13], E. Kelly Callahan[13], Scott Mcpherson[14], Emma M. Harding-Esch[1], David Macleod[15], Michaela Kelly[16], Tom Millar[16], Victor Hu[1], Paul Courtright[17], Matthew Burton[1]

1 Clinical Research Department, International Centre for Eye Health, London School of Hygiene & Tropical Medicine, London, United Kingdom, 2 Eyu-Ethiopia, Bahir Dar, Ethiopia, 3 Department of Ophthalmology, College of Medicine and Health Science, Bahir Dar University, Bahir Dar, Ethiopia, 4 Department of Ophthalmology, University of Jos, Jos, Nigeria, 5 Sightsavers, Nigeria Country Office, Abuja, Nigeria, 6 Ministry of Health, Kampala, Uganda, 7 Moroto Regional Referral Hospital, Moroto, Uganda, 8 Disease Prevention and Control Directorate, Ministry of Health, Addis Ababa, Ethiopia, 9 Fred Hollows Foundation, Alexandria, New South Wales, Australia, 10 RTI International, Kampala, Uganda, 11 International Trachoma Initiative, Decatur, Georgia, United States of America, 12 The Carter Center, Addis Ababa, Ethiopia, 13 The Carter Center, Atlanta, Georgia, United States of America, 14 RTI International, Durham, North Carolina, United States of America, 15 Department of Infectious Disease Epidemiology, London School of Hygiene & Tropical Medicine, London, United Kingdom, 16 Sightsavers International, Haywards Heath, United Kingdom, 17 Kilimanjaro Centre for Community Ophthalmology, University of Cape Town, Cape Town, South Africa

* Esmael.Ali@lshtm.ac.uk

## Abstract

There has been discussion regarding the definition of the clinical sign trachomatous trichiasis (TT) for the purposes of determining elimination of trachoma as a public health problem, and whether the definition should include the presence of trachomatous scarring (TS). A multi-country observational study was conducted in Ethiopia, Uganda and Nigeria to assess whether TS grading by field graders using the WHO simplified system in trachoma surveys are comparable with expert grading of tarsal conjunctival scarring (TCS) using a detailed system. The primary outcome was the proportion of eyes graded as "No TS" in the surveys but with TCS from expert photographic grading (the negative predictive value, NPV). In Ethiopia, Uganda and Nigeria, 545 (438 trichiasis cases and 107 comparisons), 256 (156 trichiasis cases and 100 comparisons), and 468 (352 trichiasis cases and 116 comparisons) participants, respectively, were enrolled. In Ethiopia, among 111 trichiatic eyes graded "No TS" in the surveys, 103 (92.8%) had TCS in expert photo grading, NPV 7.2% (95% CI 3.2%–13.7%). In Uganda, among 28 trichiatic eyes graded "No TS" in the surveys, 19 (67.9%) had TCS in expert photo grading, NPV 32.1% (95% CI 15.9%–52.4%). In Nigeria, among 111 trichiatic eyes graded "No TS" in the surveys, 100 (90.1%) had TCS in expert photo grading, NPV 9.9% (95% CI 5.0%–17.0%). Across settings, among eyes misdiagnosed as "No TS" in the survey, 174/250 (69.6%) had extensive

**Data availability statement:** The Amhara Public Health Institute Research Ethics Review Committee requires that all data sharing requests are reviewed and approved by them before data can be shared. Data is available to any researcher under reasonable request. To facilitate the data access process please contact ethics@lshtm.ac.uk.

**Funding:** This study is funded by Sightsavers International, UK. EH is a Wellcome International Intermediate Fellowship Fellow [Grant Number 221991/Z/20/Z], supported by the National Institute for Health Research (NIHR) (using the UK's Official Development Assistance (ODA) Funding) and Wellcome under the DHSC-Wellcome Partnership for Global Health.

**Competing interests:** I have read the journal's policy and the authors of this manuscript have the following competing interests: Michaela Kelly, Caleb Mpyet and Tom Millar are the employee of Sightsavers International, UK. Emma Harding-Esch and Ana Bakhtiari receive salary support from the International Trachoma Initiative, which receives an operating budget and research funding from Pfizer Inc., the manufacturers of Zithromax® (azithromycin). Emma Harding-Esch, Ana Bakhtiari and Caleb Mpyet are members of the core Tropical Data team, and Caleb Mpyet and Gilbert Baayenda are Tropical Data principal trainers. No conflicting relationship exists for any other author. This does not alter our adherence to PLOS ONE policies on sharing data and materials.

TCS (patches of scarring occupying ≥1/3 of the upper tarsal conjunctiva). Trichiatic eyes with TCS had more severe entropion, trichiasis, conjunctival inflammation, and corneal opacity than those without TCS. In all three settings, including TS to define a trichiasis "trachomatous" in a survey could result in the underestimation of the burden of TT. However, TCS can be effectively used to determine TT severity and management.

## Introduction

The central pathophysiological mechanism of trachomatous trichiasis (TT), the blinding stage of trachoma, is the development of scarring of the conjunctiva or deeper tissues of the eyelid. Recurrent and severe infection and chronic inflammation of the tarsal conjunctiva triggered by *Chlamydia trachomatis* leads to immuno-pathologically mediated conjunctival scarring [1]. The severity of the scarring might vary from a few dispersed white spots of scars to a severe distorting scar tissue. This fibrosis distorts and shortens the eyelid through time, resulting in inward rotation of the eyelid margin (entropion), and in-turned eyelashes touching the eyeball (trichiasis) [1]. However, the World Health Organization (WHO) definition for TT does not include a reference to the presence of scarring, although some argue that it is implicit [2].

There has been a discussion among the global trachoma expert community regarding the definition of TT for the purposes of trachoma "elimination as a public health problem", whether it should include a specific reference to and dependency on the presence of "trachomatous scarring" (TS). This was discussed at two meetings organised by WHO. In the 2014 Technical Consultation on Trachoma Surveillance by the WHO Strategic and Technical Advisory Group on NTDs, it was noted that trichiasis can be caused by scarring not involving the tarsal conjunctiva such as in epiblepharon and distichiasis, resulting in misdiagnosis as TT in trachoma-endemic settings [3]. It was then recommended that trachoma surveys should assess for and record the presence or absence of conjunctival scar in eyes with TT [3]. It was further indicated that "inability to evert the lid because of lid tightness, should be taken to indicate that the trichiasis is TT" [3]. On this basis, trichiasis cases which are not found to have TS or with eyelid tightness related to difficulty to evert are presumed to be non-trachomatous in origin. This same topic was again discussed in 2015 at the Second Global Scientific Meeting on Trachomatous Trichiasis underpinning the inadequacy of evidence to redefine TT based on either the presence of TS or the inability to evert the eyelid due to eyelid tightness [4]. This led to the recommendation that further evidence is needed to determine the best way to differentiate trichiasis due to trachoma or other causes [4].

The recommendation from the two WHO meetings led to the collection of TS data in Global Trachoma Mapping Project prevalence surveys [5] in 2015, and subsequently for surveys supported by its successor, Tropical Data (TD) [6], from all eyes identified with trichiasis to help determine if TS should be used to diagnose trichiasis

"trachomatous". GTMP supported, and TD supports, health ministries worldwide to conduct population-based prevalence surveys of active trachoma and TT by non-expert field personnel, using methodologies that conform to WHO recommendations [5,6]. TS in GTMP and TD-supported surveys is diagnosed based on the WHO simplified grading system, [7,8] "scarring easily visible as white lines, bands or sheets" in the upper tarsal conjunctiva, in which moderate and severe scars are considered "easily visible" and mild scar is excluded [9]. If the eyelid cannot be everted, it is recorded as "not able to grade" [10]. The simplified grading system was developed for use by non-specialist health personnel working in the community. The GTMP and TD training for graders, until its most recent update in 2023, [6] started with introducing the WHO simplified grading system for trachomatous inflammation—follicular (TF), trachomatous inflammation—intense (TI), TT, and TS in a presentation supported by pictures. This was followed by practising slide sets and Intergrader Agreement (IGA) test for TF on 50 photographs. Those scoring a kappa of ≥0.7 progressed to the practical in field training, which involved a field-based IGA test for TF on 50 children. Those that score kappa of ≥0.7 for TF qualified to participate as graders in the survey [5,6,10–12].

Many trachoma-endemic countries are in the process of preparing data for dossiers to validate the elimination of trachoma as a public health problem for submission to WHO [13]. The data presented in these dossiers should demonstrate that each previously endemic district has reached the elimination prevalence threshold for TT unknown to the health system (<0.2% in adults aged ≥15 years). This requires reliable data on the residual prevalence of TT. However, it remained uncertain whether the diagnosis of TT for this purpose should include the simultaneous presence of TS. In parallel to the data being collected as part of TD-supported surveys, it has been proposed to generate evidence from various settings on the inclusion of TS in the TT definition based on survey gradings by non-expert field personnel.

We conducted a multi-country study to (a) estimate the proportion of trichiasis cases identified during surveys in trachoma-endemic regions that were originally recorded as having "no TS" by non-expert field personnel using the WHO simplified grading system that actually have a degree of tarsal conjunctival scarring (TCS) by expert re-grading of photographs using a detailed grading system, (b) compare the phenotype of trichiasis cases with and without TCS; and (c) identify factors associated with TCS severity among trichiasis cases in trachoma-endemic settings. We used the term "TS" to refer to the non-specialist survey grading as per the simplified WHO definition for "trachomatous scarring" (the presence of easily visible scars in the upper tarsal conjunctiva) [7] while the term "Tarsal Conjunctival Scarring" or "TCS" was used to refer to the results of expert grading using a detailed grading system [14] developed modifying the WHO FPC grading system [15].

## Materials and methods

### Ethics statement

This study was approved by the Amhara Public Health Institute Research Ethics Review Committee (ref # 02/100/2018), and Benishangul Gumuz (BG) Regional Health Research Ethics Review Committee, (ref # 10/2/12) in Ethiopia; Vector Control Division-Research & Ethics Committee (Ref # VCDREC/097) in Uganda; National Health Research Ethics Committee of Nigeria (Ref # NHREC/01/01/2007-09/8/2018C and Ref # NHREC/01/01/2007-17/03/2021C), London School of Hygiene & Tropical Medicine Ethics Committee (ref # 15496), and Emory University Institutional Review Board (ref # IRB00103492). Written informed consent was obtained prior to enrolment from participants in all settings. If a participant was unable to read and write, the information sheet and consent form were read to them and their consent recorded by thumbprint, in the presence of a witness.

### Study design and participants

A multi-country comparative cross-sectional study was conducted in settings with varying trichiasis burden, in Ethiopia (two regions), Uganda and Nigeria, to compare the TS grading (refers to the WHO simplified tarsal conjunctival scarring sign) collected from trichiasis cases by non-expert field graders in trachoma population-based surveys, with expert

 

grading using a detailed TCS grading system [14]. As of April 2024, Ethiopia, Nigeria and Uganda had 774, 197 and 52 districts with prevalence of trichiasis unknown to health system ≥0.2% in ≥15-year-olds, respectively [16].

Participants were un-operated trichiasis cases, defined as those having one or more eyelash touching the eyeball or evidence of epilation of the upper eyelid in either or both eyes ('cases' hereafter), with accompanying TS data, identified from trachoma surveys. These cases were re-examined in their villages by an independent team of experts. Trichiasis cases with no TS data, evidence or history of trichiasis surgery in the study eye before or after the TD-supported trachoma surveys, < 15 years of age, and who refused for their tarsal conjunctiva photograph to be taken were excluded. As a comparison for the cases, frequency-matched (age, sex, and location) trichiasis-free individuals, defined as people without a history or evidence of trichiasis (including epilation) or trichiasis surgery, who were identified from the same trachoma survey population as the cases were randomly selected and enrolled ('comparisons' hereafter). The use of 'comparisons' ensured a case mix for the photographic grading and reduced potential bias in the photograph grading in favour of TS if it was thought that all the photographs were from people with trichiasis. People identified as having no trichiasis, or no trichiasis surgery during the trachoma survey, but who were found to have trichiasis or evidence or history of epilation or trichiasis surgery in either eye during the expert re-examination, were excluded from the comparisons.

**Recruitment process**

The recruitment for this study was conducted between August 2018 and August 2021 approximately six-months after the completion of a trachoma survey that documented TS data in trichiasis cases. Data collection in Ethiopia took place from August 18 to September 22, 2018, in the BG region, and from September 26 to November 8, 2018, in the Amhara region. In Uganda, data collection was conducted from November 1, 2018, to March 19, 2019. Data collection in Nigeria took place from June 17, 2019, to August 16, 2021, with interruptions due to delays in trachoma surveys, security concerns, and the COVID-19 pandemic. Trichiasis cases were first identified from the list of all people examined during trachoma surveys. In Amhara region, Ethiopia, 400 cases were randomly selected for enrolment into this study from a total of 935 trichiasis patients. Individuals who were neither available during the recruitment period nor eligible were replaced by other location-, sex-, and age-matched trichiasis cases selected randomly from the same list of people examined during the trachoma survey. In BG region of Ethiopia, Uganda, and Nigeria where the number of identified trichiasis cases in the most recent trachoma survey was less than 400, the research team accessed and examined all available trichiasis cases.

Comparisons were randomly selected from a list of individuals without trichiasis, frequency matched to cases by district, sex, and within a ten-year age range. Initial matching was performed in Stata by randomly pairing potential comparisons to trichiasis cases based on district and age group (within a 10-year interval). To ensure comparable sex distribution between cases and comparisons, a proportionate number of female and male comparisons were randomly selected in Stata, mirroring the sex distribution of the cases. For example, if 60% of cases were female, 60% of comparisons were also randomly selected as female and 40% as male. If a selected comparison was unavailable, declined participation, or was ineligible, a replacement with the same sex, from the same village and within the same 10-year age range as the case, was randomly selected from the comparison list. If no eligible replacement was available in the same village, the nearest village was used following the same procedure.

The list of cases and comparisons with their demographic details, household head and locations was communicated to the local village leaders and community health workers with the date and place of recruitment to the study. The research team gave transport services to and from the recruitment site, a health facility, or a central location within the villages. The research team travelled to the houses of those that were not able to come to the recruitment site. The potential participants were asked for their full demographic details and their household head name to confirm identity at first encounter, which was compared against the list to confirm that the correct person had been identified. A GPS device (Garmin GPS reader) was used (using the coordinates collected in the trachoma surveys) to locate the houses of people that could not be found using their demographic details.

## Assessments

A brief demographic questionnaire was employed by the study team to record name, sex, age, address, ethnicity, and literacy for all participants. For trichiasis cases, a brief questionnaire was introduced to elicit information on the duration of trichiasis, history of epilation, and other trichiasis-related concerns. Visual acuity was measured using PeekAcuity [17]. Five independent observers (three in Ethiopia and one each in Nigeria and Uganda), masked to the TS grading status of the trichiasis cases determined in the trachoma surveys, examined the participants' eyes using magnifying loupes and a torch.

The detailed WHO FPC grading system [15] with additional assessment of TCS, degree of entropion, type, location and number of trichiatic lashes was employed to determine cicatricial trachoma phenotype. Eyelashes touching the eyeball were counted and subdivided by the part of the eyeball contacted: cornea, lateral, or medial conjunctiva. Trichiasis subtypes were recorded as metaplastic, misdirected, and entropic [18]. Clinical evidence of epilation was identified by broken or newly growing lashes, or areas of absent eyelashes. Upper eyelid entropion was graded by assessing the degree of eyelid margin inward rotation [18]. Corneal scarring was graded using a previously described detailed system [19]. Tarsal conjunctival inflammation, characterised by papillary hypertrophy and diffusive infiltration, was graded as 'absent', 'minimal', 'moderate', 'severe' (pronounced) based on the classification in the FPC grading system [15]. A detailed TCS grading system developed by Hu *et al*. modifying the WHO FPC grading to provide objective definitions for "mild" and "moderate" scarring was employed to determine the presence and severity of TCS, Table 1 [14,20]. Digital photographs of the upper tarsal conjunctiva were taken with the upper eyelid everted from both the cases and the comparisons using Nikon D90 digital SLR camera with 105 mm macro lens and R1C1 flash units. Both text and photography data were collected electronically and stored in the London School of Hygiene and Tropical Medicine (LSHTM) servers through the Open Data Kit (ODK) system.

The data collection team was composed of an examiner, coordinator, photographer, consent and data collectors (2) and examination assistant. The examiners and photographers from Ethiopia, Uganda and Nigeria were trained and standardised in Ethiopia between June 24 and 29, 2018 by VH. The training included practical patient-based training on key outcome measures such as TCS and inflammation, trichiasis, entropion, and corneal opacity, and photographic-based training on TCS. The examiners were standardised against the gradings of VH on 50 live participants. The Ethiopia and Nigeria graders scored >0.70 kappa during the interobserver agreement test, while the Uganda examiner was replaced by AG, who was trained and standardised later in Ethiopia against EH gradings and achieved the score of >0.70 kappa. The rest of the research team were trained on their roles in each study setting separately based on the project Standard Operating Procedure (SOP).

## Photograph grading

The photographs of the tarsal conjunctiva of the comparisons were mixed in with the trichiasis cases' conjunctival photographs for grading to avoid bias in grading. Independent experts, masked to the previously graded TS status of the cases, graded all the photographs using both the new detailed grading system [14] and the WHO FPC scarring grading system [15]. The photos from the Ethiopia component were graded by a senior consultant ophthalmologist (MJB), while the photos from Uganda and Nigeria were graded by two experienced and certified TD survey graders. At first, four graders were trained and standardised on the new grading system by EH using photos graded by MJB. These individuals' independent grading of 50 photos was assessed against MJB's grading of the photos. Among the four, the two that obtained a kappa score >0.70 independently graded all the photos from Uganda and Nigeria. Each photo was independently graded by two graders. In cases of disagreement, the discrepant gradings were adjudicated by MJB. Therefore, the final grading used for analysis was the consensus between the two independent TD survey graders or, when discrepant, the adjudicated grade provided by MJB.

**Table 1. Tarsal conjunctival scarring grading system used in this study.**

**(a) The World Health Organization (WHO) FPC conjunctival scarring grading system [15]**

**Conjunctival scarring (C)**

| | |
|---|---|
| C 0 | No scarring on the conjunctiva |
| C 1 | Mild: fine scattered scars on the upper tarsal conjunctiva, or scars on other parts of the conjunctiva. |
| C 2 | Moderate: more severe scarring but without shortening or distortion of the upper tarsus. |
| C 3 | Severe: scarring with distortion of the upper tarsus. |

***(b) The detailed tarsal conjunctival scarring grading system, developed by Hu et al., 2011 [14] modifying the FPC conjunctival scarring grading system***

| Grade | Definition* |
|---|---|
| S0 | No scar |
| S1 | Scarring occupying <1/3 of the upper eyelid‡ |
| S1a | One or more pinpoint scars and/or a single line of scaring < 2mm in length† |
| S1b | Multiple lines of scarring <2mm in length |
| S1c | One or more lines/patches of scarring each 2mm or more in length/maximal dimension |
| S2 | Patches of scarring occupying in surface area ≥1/3 but <2/3 of the upper eyelid |
| S3 | Patches of scarring occupying in surface area ≥2/3 of the upper eyelid and/or distortion |
| Ungradeable | Not possible to evert adequately or poor photo quality such as from blurring or conjunctival blanching |

‡Upper eyelid refers to the portion of the tarsal conjunctiva outlined by the dotted lines in S1 Fig.

†2mm is the approximate width of the lower lid margin, which is readily available for comparison.

## Sample size

The sample size was based around the primary aim of determining the proportion of people with trichiasis found in a survey who were at that time said to have no TS based on the simplified grading system, who in fact did have some degree of scarring as assessed by experts using the new grading system. We made the following assumptions: (1) in the original surveys approximately 25% of people with trichiasis were graded as not having TS; (2) of the people graded as trichiasis without TS, 50% were false negatives, and will in fact be found to have some degree of conjunctival scarring on the photograph grading; (3) we require at least + /- 10% precision on this proportion originally graded as false negatives. Using these assumptions, a sample of 400 people per site with trichiasis identified in a recent survey would be expected to have around 100 people graded as trichiasis without scarring. This will be sufficient to provide + /-10% precision in the estimate of the proportion of false negatives for TS. An additional 100 people without trichiasis were recruited, examined, and photographed, to reduce systematic bias in the photograph grading.

## Statistical analysis

Data collected using ODK were pulled from the server at LSHTM and transferred to Stata version 17.0 for analysis. The analysis was focused on (a) comparing the TS data collected from trichiasis cases by non-expert field graders in trachoma population-based surveys with an independent expert grading in the field and using high-resolution digital photographs

collected during the field grading, and (b) comparing the phenotypes of trichiasis with and without TCS. Analysis was conducted separately for each country. The unit of analysis was eyes, unless specified otherwise.

The primary analysis was the determination of the proportion of people with trichiasis who were originally graded in the surveys to have "no TS", but who had some degree of conjunctival scarring from the photographic grading: the false negative rate (negative predictive value, NPV). All eyes that were diagnosed as having trichiasis in the survey field grading, regardless of the presence or absence of trichiasis in the detailed field grading, were included in the analysis. The TCS grading used for the primary outcome analysis and most secondary outcome analyses (unless specified) was the detailed grading system developed by Hu *et al* [14]. Secondary analyses included the estimation of the sensitivity, specificity, and positive predictive value (PPV) of the initial survey grading compared to the photograph grading. The relationship between the degree of conjunctival scarring (determined by the photograph grading) and key clinical characteristics that may be associated with TCS and TCS severity such as the nature (entropion, metaplastic, misdirected) and severity of the trichiasis (numbers of eyelashes) were investigated in ordinal logistics regression, first in a univariable model and then those with a p-value < 0.2 were included in the final multivariable model. Age and sex were included in the model as fixed variables to control for potential confounding effects. Eye-level analyses were adjusted for between eye correlation using robust standard errors clustered at the participant level to account for correlation between eyes from the same individual. Sensitivity analysis was conducted by repeating the key analyses using only one eye per individual. The analysis of data from comparison participants was descriptive; no inferential statistical analyses were conducted.

## Results

### Participant enrolment

In Ethiopia 545 participants, 438 cases and 110 comparisons, were enrolled. In Amhara, among 935 people with un-operated trichiasis and TS data identified from the most recent survey conducted in June 2018 in the Eastern part of the region, 400 were randomly selected. The cases that met and did not meet the eligibility criteria are indicated in Fig 1a. In addition, 100 comparisons were enrolled. In BG, 38 cases and 10 comparisons were enrolled from surveys conducted March 2018, Fig 1b.

In Uganda, 337 people with un-operated trichiasis and TS data were identified from the most recent trachoma surveys conducted between December 2017 and September 2018 from Karamoja, West Nile and Busoga regions, (Fig 1c), among which 156 that met the inclusion criteria, and 100 comparisons were enrolled. This resulted in 256 participants being enrolled, 156 trichiasis cases and 100 comparisons.

in Nigeria 912 people with un-operated trichiasis and TS data were identified from trachoma surveys conducted between November 2018 and June 2021 in four states (Bauchi, Jigawa, Kano, Katsina). Among these, 355 that met the inclusion criteria were enrolled and 352 had complete data (Fig 1d). In addition, 116 comparisons were enrolled. Complete data were collected from 468 participants: 352 trichiasis cases and 116 comparisons.

### Demographic and clinical characteristics

The demographic characteristics of trichiasis cases and comparisons for each study site are presented in Table 2. Females and participants with no formal education constituted the majority of the participants in all settings. Most of the participants in Ethiopia were farmers and married. In Uganda, the majority of cases did not have a job (74.6%) and were widowed (52.3%). There was no evidence of a difference in terms of age or sex in trichiasis cases assessed for eligibility compared to those not found in Uganda and Nigeria.

The clinical characteristics of eyes of the cases and comparisons enrolled in the study from each study site are presented in Table 3. In Ethiopia, among 594 eyes diagnosed with unoperated trichiasis in the trachoma surveys, 120 (20.2%) had neither trichiasis nor evidence of epilation in the detailed field grading. More than one-third of the eyes had minor trichiasis (≤5 eyelashes touching the eye or evidence of epilation in <1/3$^{rd}$ of the eyelid) with 225 (37.9%) eyes

Amhara Region, Ethiopia

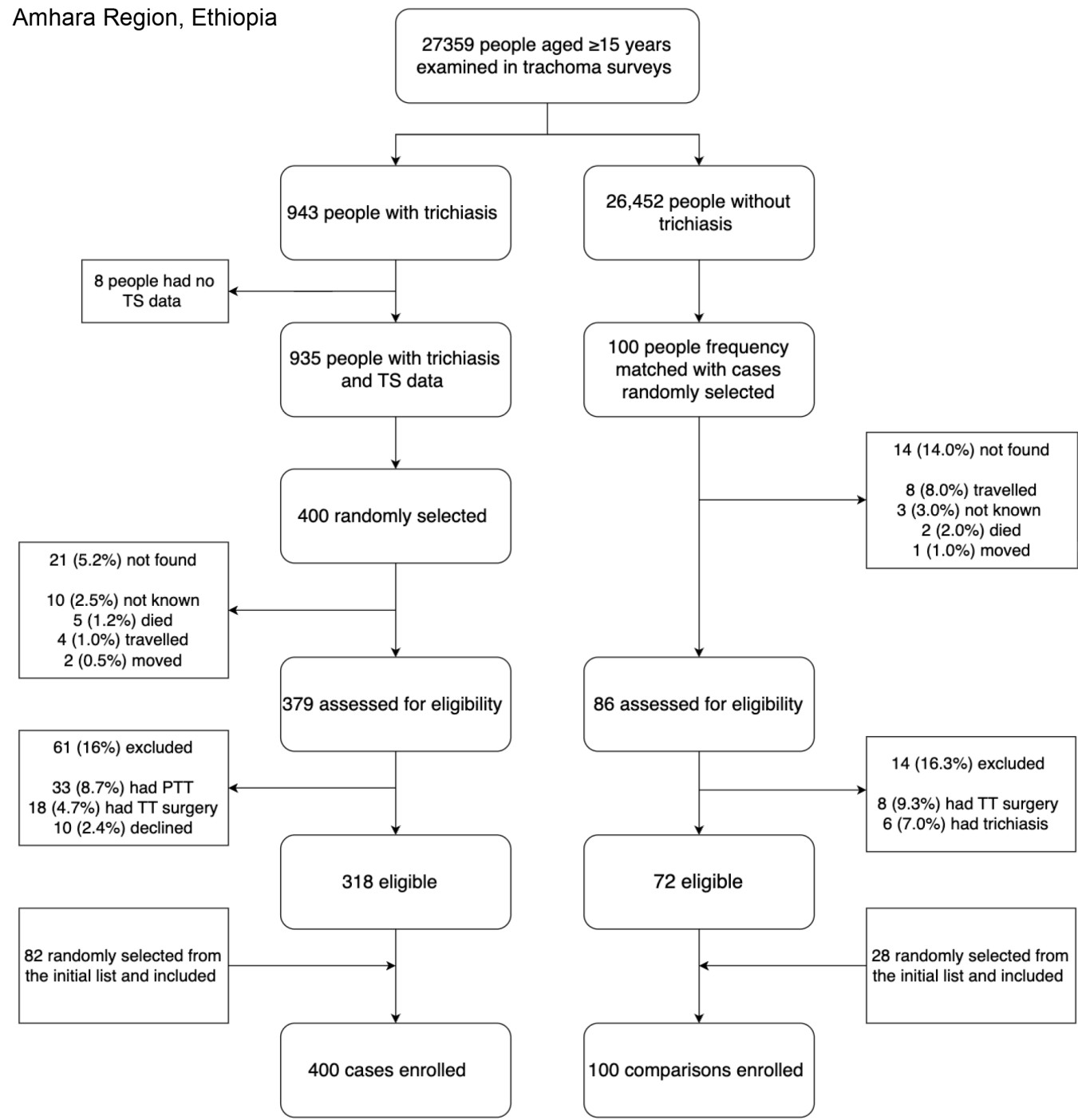

Note: TT=Trachomatous trichiasis, PTT=Post operative Trachomatous trichiasis, TS = Trachomatous Scarring (WHO Simplified System)

Benishangul Gumuz (BG) Region, Ethiopia

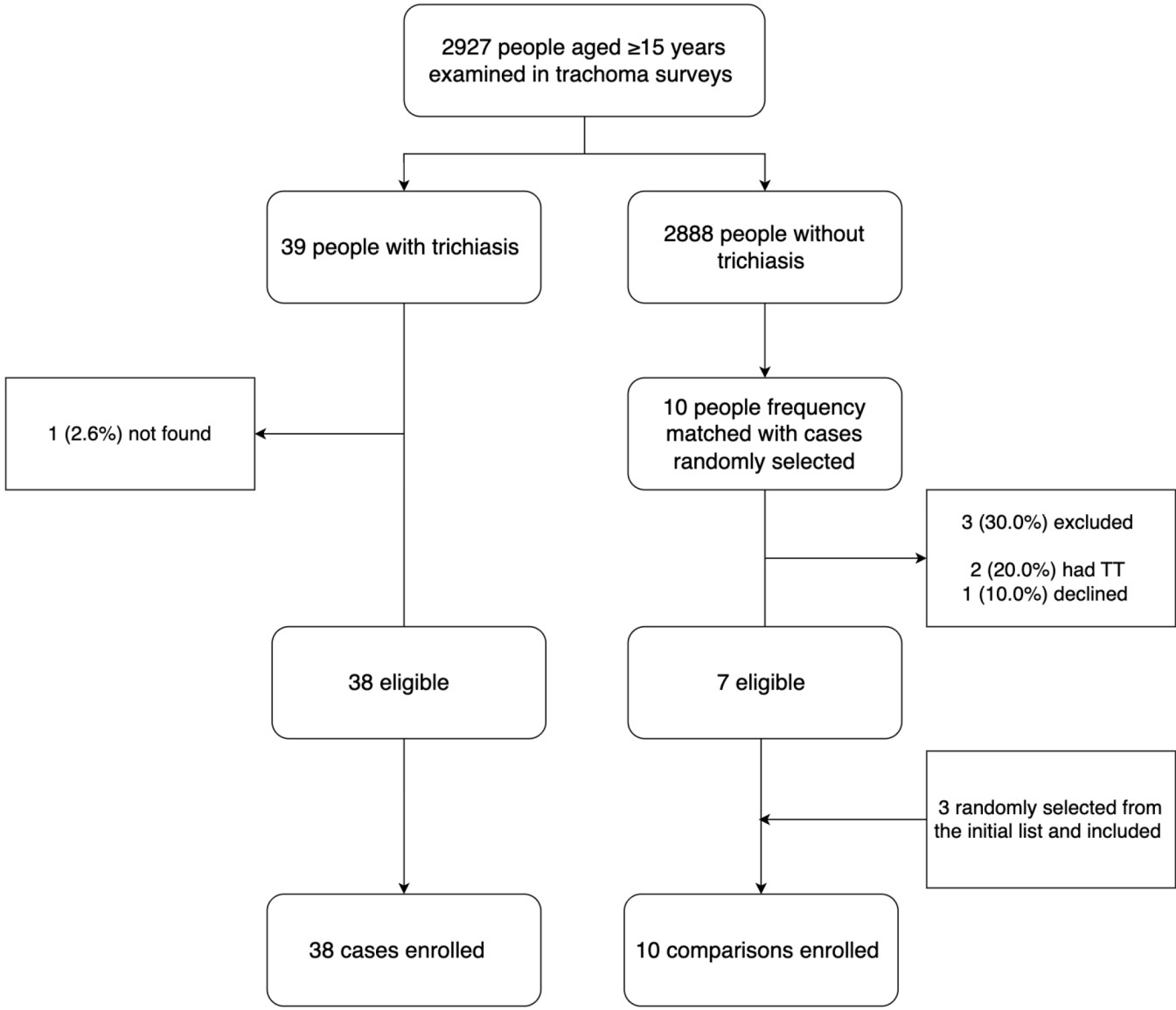

Note: *TT=Trachomatous trichiasis, PTT=Post operative Trachomatous trichiasis, TS = Trachomatous Scarring (WHO Simplified System)*

Uganda

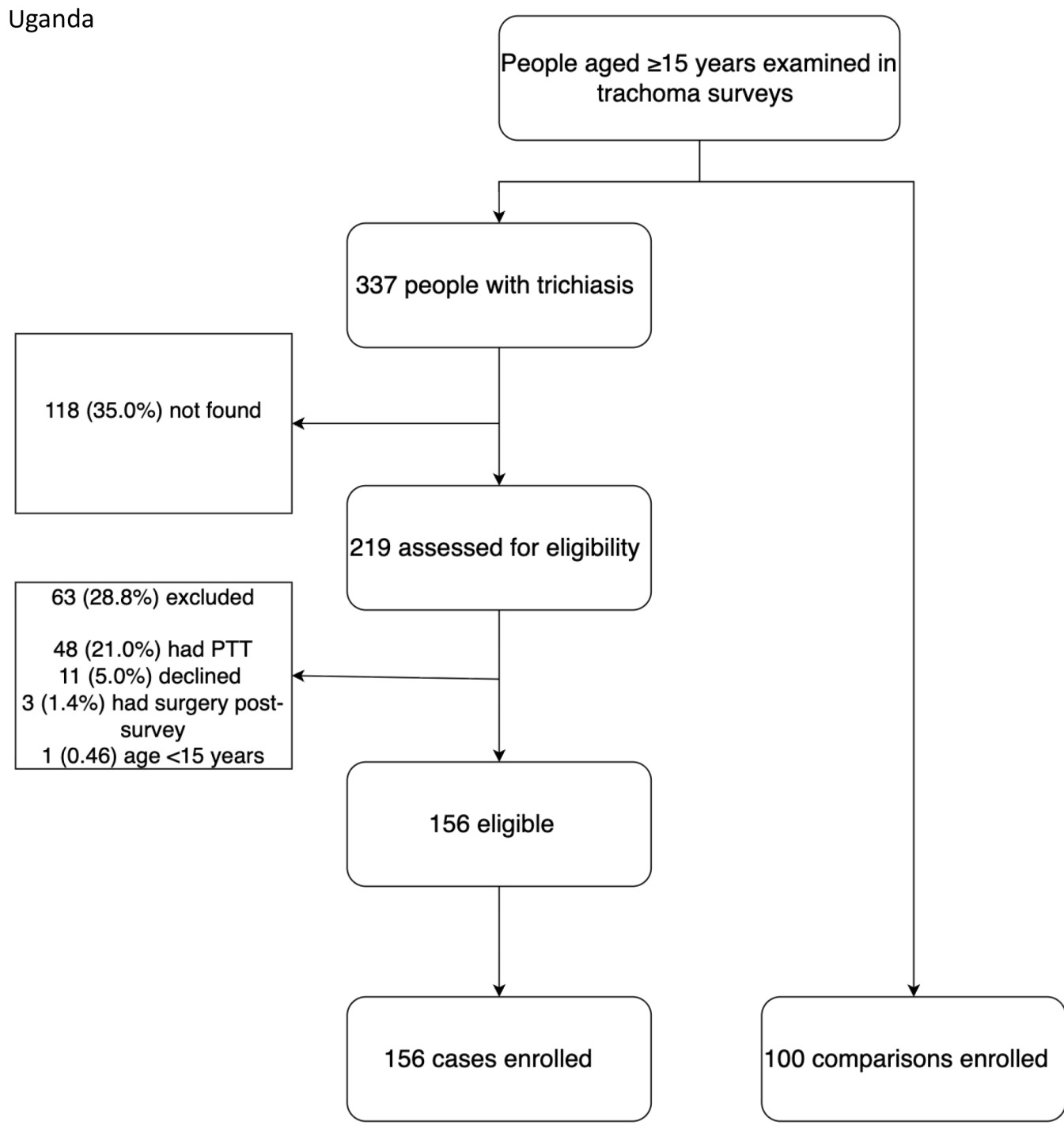

Note: TT=Trachomatous trichiasis, PTT=Post operative Trachomatous trichiasis, TS = Trachomatous Scarring (WHO Simplified System)

PLOS Global Public Health

Nigeria

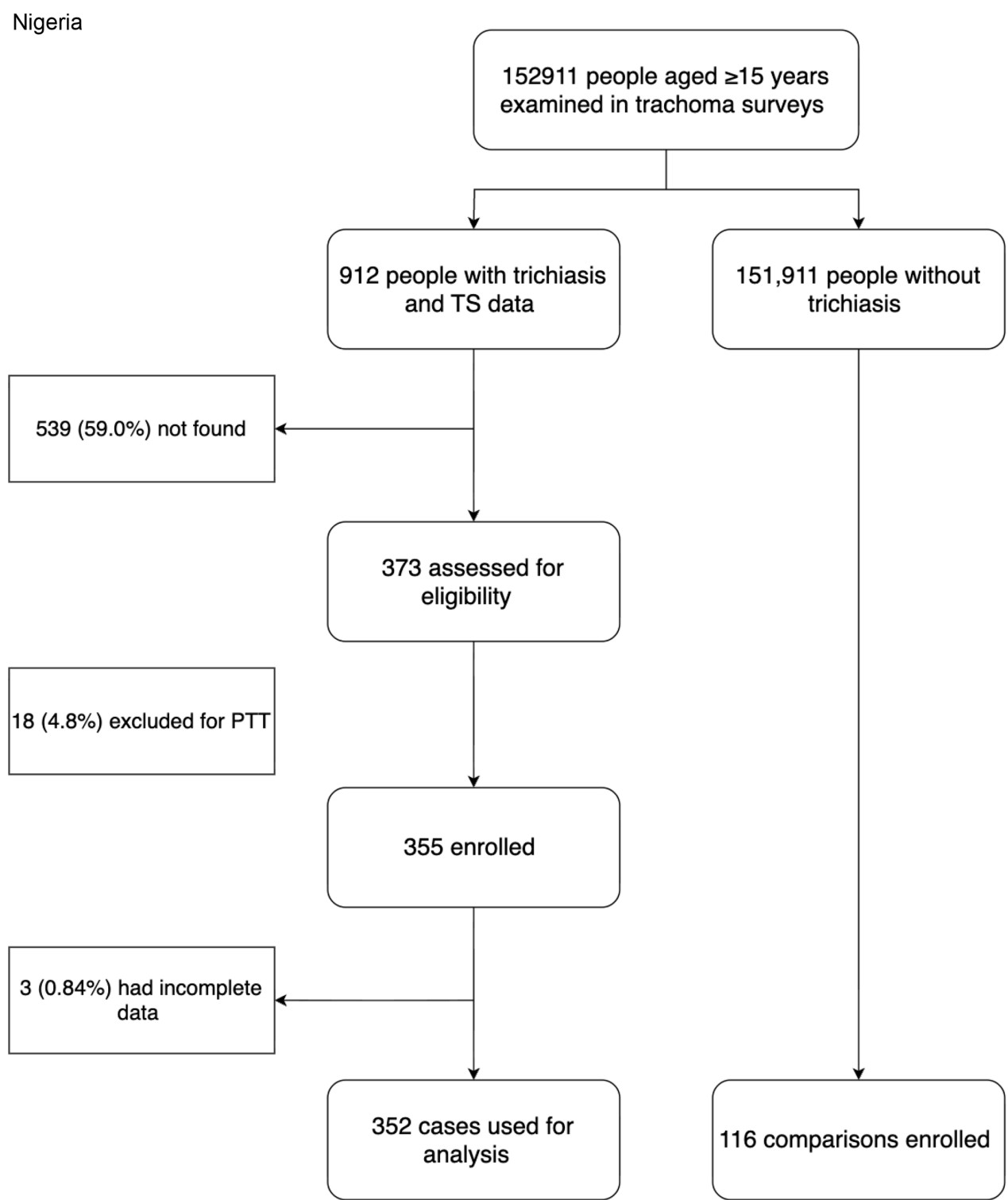

Note: *TT=Trachomatous trichiasis, PTT=Post operative Trachomatous trichiasis, TS = Trachomatous Scarring (WHO Simplified System)*

**Fig 1. Recruitment history of trichiasis cases and comparisons without trichiasis in Ethiopia, Uganda and Nigeria.**

Table 2. Demographic characteristics of trichiasis cases and comparisons, by study site.

| Variables | Ethiopia | | | | Uganda | | | | Nigeria | | | |
|---|---|---|---|---|---|---|---|---|---|---|---|---|
| | Cases | | Comparisons | | Cases | | Comparisons | | Cases | | Comparisons | |
| | n/438 | (%) | n/110 | (%) | n/156 | (%) | n/100 | (%) | n/352 | (%) | n/116 | (%) |
| **Age in years, mean (SD)** | 54.3 | (16.6) | 53.4 | (15.7) | 58.1 | (20.3) | 48.5 | (16.6) | 56.4 | (16.9) | 51.4 | (16.2) |
| **Sex, female** | 326 | (74.4) | 84 | (76.4) | 127 | (81.4) | 60 | (60.0) | 232 | (65.9) | 75 | (64.7) |
| **Education status** | | | | | | | | | | | | |
| No education | 390 | (89.0) | 94 | (85.5) | *136* | *(87.7)* | *49* | *(49.0)* | 163 | (46.3) | 34 | (19.3) |
| Religious or other traditional education | 11 | (2.5) | 1 | (0.9) | 3 | (1.9) | 10 | (10.0) | 127 | (36.1) | 43 | (37.1) |
| Primary- secondary school | 36 | (8.2) | 15 | (13.6) | 16 | (10.3) | 39 | (39.0) | 56 | (15.9) | 29 | (25.0) |
| Higher education | 1 | (0.2) | 0 | (0.0) | 0 | (0.0) | 2 | (2.0) | 6 | (1.7) | 10 | (8.6) |
| **Job** | | | | | | | | | | | | |
| No job | 73 | (36.7) | 12 | (10.9) | 116 | (74.6) | 44 | (44.0) | 114 | (32.4) | 31 | (26.7) |
| Farmer | 338 | (77.2) | 93 | (84.6) | *37* | *(23.9)* | *40* | *(40.0)* | *122* | *(34.7)* | *31* | *(26.7)* |
| Employed/self employed | 19 | (4.3) | 5 | (345) | 2 | (1.3) | 14 | (14.0) | 114 | (32.4) | 54 | (46.6) |
| Daily labourer | 8 | (1.8) | 0 | (0.0) | 0 | (0.0) | 2 | (2.0) | 2 | (0.57) | 0 | (0.0) |
| **Marital status** | | | | | | | | | | | | |
| Single | 20 | 4.6) | 2 | (1.8) | *11* | *(7.1)* | *16* | *(16.0)* | 2 | (0.57) | 5 | (4.3) |
| Married | 261 | (59.6) | 77 | (70.0) | 50 | (32.3) | 52 | (52.0) | 326 | (92.6) | 107 | (92.2) |
| Divorced | 47 | (10.7) | 10 | (9.1) | 13 | (8.4) | 2 | (2.0) | 1 | (0.28) | 0 | (0.0) |
| Widowed | 110 | (25.1) | 21 | (19.1) | 81 | (52.3) | 30 | (30.0) | 23 | (6.5) | 4 | (3.5) |
| **Weight in kilograms, mean (SD)** | 48.1 | (8.2) | 50.9 | (7.7) | 54.6 | (10.6) | 60.7 | (11.1) | 54.8 | (16.1) | 56.2 | (14.9) |
| **Height in centimetres, mean (SD)** | 156.7 | (8.1) | 158.0 | (7.2) | 161.6 | (7.7) | 165.4 | (9.0) | 156.6 | (17.4) | 159.7 | (9.4) |
| **BMI, mean (SD)** | 15.3 | (7.6) | 16.1 | (2.1) | 16.9 | (10.5) | 18.3 | (3.1) | 18.8 | (16.1) | 17.6 | (4.6) |

The weight, height and BMI (Body Mass Index) data for Ethiopia, Uganda, and Nigeria are from 436, 148, and 345 cases, respectively.

having one or two eyelashes touching the eyeball. In Uganda, among 223 eyes diagnosed with unoperated trichiasis in the trachoma surveys, 71 (31.8%) had neither trichiasis nor evidence of epilation in the detailed field grading, and 50.0% had minor trichiasis with 38 (17.0%) eyes having one or two eyelashes touching the eyeball. In Nigeria, among 507 eyes diagnosed with unoperated trichiasis in the trachoma surveys, 206 (40.6%) had neither trichiasis nor evidence of epilation in the detailed field grading, and 161 (53.5%) had minor trichiasis, with 106 (20.9%) eyes having one or two eyelashes toughing the eyeball. There was more entropion, lower eyelid trichiasis and tarsal conjunctival inflammation in the case eyes than the comparison eyes in all the three settings.

## Tarsal conjunctival scarring frequency

The TS grading in the trachoma survey, and the TCS grading in the detailed field grading and photographic grading are presented in Table 4, for trichiasis eyes identified in the trachoma survey and their comparisons. During the trachoma survey, in the 1322 eyes identified with trichiasis and with TS data, 1058 eyes (80.0%) were diagnosed with TS (Ethiopia 81.3%, Uganda 86.1%, Nigeria 75.9%). Of the 1322 eyes, when 1297 gradable eyes were regraded using the detailed field grading, 1181 (91.1%) were diagnosed with TCS (Ethiopia 95.1%, Uganda 88.8%, Nigeria 87.1%). Using the expert photographic grading, the proportions were slightly higher, with 1153 (91.9%) of 1255 eyes with gradable photos diagnosed with TCS (Ethiopia 97.3%, Uganda 74.9%, Nigeria 92.8%). The expert photographic grading also showed that 47.2% of the comparison participants' eyes (Ethiopia 72.1%, Uganda 13.9%, Nigeria 51.9%) had TCS (no TS data available from the survey results, as TS only looked for in individuals recorded as having trichiasis).

**Table 3. Clinical characteristics of eyes diagnosed with trichiasis in trachoma surveys and their comparisons, by study site.**

| Characteristics | All three countries | | Ethiopia | | Uganda | | Nigeria | |
|---|---|---|---|---|---|---|---|---|
| | Trichiasis case eyes N=1324[a] (%) | Compari-son eyes N=652 (%) | Trichiasis case eyes N=594 (%) | Compari-son eyes N=220 (%) | Trichiasis case eyes N=223 (%) | Compari-son eyes N=200 (%) | Trichiasis case eyes N=507 (%) | Comparison eyes N=232 (%) |
| **Trichiasis (eyelash #) [b]** | | | | | | | | |
| Eyes with no trichiasis or epilation | 397 (30) | | 120 (20.2) | | 71 (31.8) | | 206 (40.6%) | |
| Epilated successfully (diag-nosed by epilation evidence) | 69 (5) | | 34 (5.7) | | 3 (1.3) | | 32 (6.3) | |
| 1 | 220 (16.6) | | 143 (24.1) | | 25 (11.2) | | 52 (10.3) | |
| 2 | 149 (11.2) | | 82 (13.8) | | 13 (5.8) | | 54 (10.6) | |
| 3 - 5 | 217 (16.4) | | 108 (18.2) | | 42 (18.8) | | 67 (13.2) | |
| 6 - 9 | 105 (7.9) | | 42 (7.7) | | 18 (8.1) | | 45 (8.9) | |
| 10 - 19 | 97 (7.3) | | 34 (5.7) | | 28 (12.6) | | 35 (6.9) | |
| 20+ | 70 (5.3) | | 31 (5.2) | | 23 (10.3) | | 16 (3.2) | |
| **Trichiasis severity[‡]** *(incudes epilation)* | | | | | | | | |
| Minor | 547 (59.0) | | 310 (65.4) | | 76 (50.0) | | 161 (53.5) | |
| Major | 380 (41.0) | | 164 (34.6) | | 76 (50.0) | | 140 (46.5) | |
| **Eyelash type[‡]** | | | | | | | | |
| Epilated successfully | 69 (7.4) | | 34 (7.2) | | 3 (2.0) | | 32 (10.6) | |
| Entropic | 292 (31.5) | | 143 (30.2) | | 59 (38.8) | | 90 (29.9) | |
| Metaplastic | 375 (40.4) | | 232 (48.9) | | 63 (41.4) | | 80 (26.6) | |
| Misdirected | 63 (6.8) | | 11 (2.3) | | 15 (9.9) | | 37 (12.3) | |
| Mixed | 128 (13.8) | | 54 (11.4) | | 12 (7.9) | | 62 (20.6) | |
| **Eyelash location[‡]** | | | | | | | | |
| Epilated successfully | 69 (7.4) | | 34 (7.2) | | 3 (2.0) | | 32 (10.6) | |
| Corneal only | 461 (49.7) | | 285 (60.1) | | 76 (50.0) | | 100 (33.2) | |
| Peripheral only | 121 (13.0) | | 51 (10.8) | | 22 (14.5) | | 48 (15.9) | |
| Mixed location | 276 (29.8) | | 104 (21.9) | | 51 (33.6) | | 121 (40.2) | |
| **Tarsal conjunctival inflammation [a]** | | | | | | | | |
| None | 429 (32.4) | 427 (65.5) | 46 (7.7) | 92 (41.8) | 57 (25.6) | 112 (56.0) | 326 (64.3) | 223 (96.1) |
| Mild | 251 (19.0) | 146 (22.4) | 124 (20.9) | 64 (29.1) | 58 (26.0) | 80 (40.0) | 69 (13.6) | 2 (0.8) |
| Moderate | 341 (25.8) | 47 (7.2) | 222 (37.4) | 39 (17.7) | 57 (25.6) | 6 (3.0) | 62 (12.2) | 2 (0.86) |
| Severe | 278 (21.0) | 30 (4.6) | 201 (33.8) | 24(10.9) | 45 (20.2) | 2 (1.0) | 32 (6.3) | 4(1.72) |
| Ungradable | 25 | 2 (0.3) | 1 (0.2) | 1 (0.4) | 6 (2.7) | 0 | 18 (3.6) | 1 (0.43) |
| **Entropion [a]** | | | | | | | | |
| None | 287 (21.7) | 444 (68.1) | 73 (12.3) | 124 (56.4) | 35 (15.7) | 118 (59.0) | 179 (35.3) | 202 (87.1) |
| Mild | 308 (23.3) | 165 (25.3) | 172 (29.0) | 81 (36.8) | 34 (15.2) | 62 (31.0) | 102 (20.1) | 22 (9.5) |
| Moderate | 483 (36.5) | 43 (6.6) | 252 (42.4) | 15 (6.8) | 86 (38.6) | 20 (10.0) | 145 (28.6) | 8 (3.5) |
| Severe | 174 (13.1) | 0 | 65 (10.9) | 0 | 64 (28.7) | 0 | 45 (8.9) | 0 |
| Complete | 72 (5.4) | 0 | 32 (5.4) | 0 | 4 (1.8) | 0 | 36 (7.1) | 0 |
| **Lower eyelid trichiasis [a]** | | | | | | | | |
| No | 1261 (9563 | 646 (996 | 556 (93.6) | 217 (98.6) | 217 (97.3) | 200 (100.0) | 488 (96.2) | 229 (98.7) |
| Yes | 63 (4.8) | (0.9) | 38 (6.4) | 3 (1.4) | 6 (2.7) | 0 | 19 (3.7) | 3 (1.3) |
| ***Corneal opacity [a]*** | | | | | | | | |
| None (CC0) | 590 (44.6) | 546 (83.7) | 154 (25.9) | 155 (70.4) | 101 (45.3) | 185 (92.5) | 335 (66.1) | 206 (88.8) |

*(Continued)*

**Table 3.** (Continued)

| Characteristics | All three countries | | Ethiopia | | Uganda | | Nigeria | |
|---|---|---|---|---|---|---|---|---|
| | Trichiasis case eyes N = 1324[a] (%) | Compari-son eyes N = 652 (%) | Trichiasis case eyes N = 594 (%) | Compari-son eyes N = 220 (%) | Trichiasis case eyes N = 223 (%) | Compari-son eyes N = 200 (%) | Trichiasis case eyes N = 507 (%) | Comparison eyes N = 232 (%) |
| Peripheral (CC1) | 171 (12.9) | 32 (4.9) | 95 (16.0) | 19 (8.6) | 26 (11.7) | 5 (2.5) | 50 (9.9) | 8 (3.4) |
| Off central faint (CC2a) | 105 (7.9) | 23 (3.5) | 91 (15.3) | 18 (8.2) | 5 (2.2) | 4 (2.0) | 9 (1.8) | 1 (0.43) |
| Off central dense (CC2b) | 18 (1.4) | 2 (0.3) | 9 (1.5) | 2 (0.9) | 3 (1.3) | 0 | 6 (1.2) | 0 (0.0) |
| Central faint (CC2c) | 319 (24.1) | 37 (5.7) | 187 (31.5) | 18 (8.2) | 69 (30.9) | 5 (2.5) | 63 (12.4) | 14 (6.0) |
| Central dense (CC2d) | 53 (4.0) | 7 (1.1) | 25 (4.2) | 5 (2.3) | 7 (3.1) | 0 | 21 (4.1) | 2 (0.86) |
| Total central dense (CC3) | 44 (3.3) | 3 (0.5) | 27 (4.6) | 2 (0.9) | 5 (2.2) | 1 (0.5) | 12 (2.4) | 0 (0.0) |
| Phthisis (CC4) | 24 (1.8) | 2 (0.3) | 6 (1.0) | 1 (0.4) | 7 (4.1) | 0 | 11 (2.2) | 1 (0.43) |
| **Vision [a]** | | | | | | | | |
| Normal vision | 654 (50.6) [a] | 453 (70.1) [b] | 294 (49.9) | 149 (68.7) [d] | 112 (52.3) | 149 (74.5) | 248 (50.6) | 155 (67.7) |
| Moderate visual impairment (<6/18 - ≥ 6/60) | 412 (31.9) | 136 (21.1) | 191 (32.4) | 53 (24.4) | 66 (30.8) | 41 (20.5) | 155 (31.6) | 42 (18.3) |
| Sever visual impairment (<6/60 - ≥ 3/60 | 47 (3.6) | 19 (2.9) | 26 (4.4) | 5 (2.3 | 1 (0.5) | 2 (1.0) | 20 (4.1) | 12 (5.5) |
| Blind (<3/60) | 180 (13.9) | 38 (3.9) | 78 (13.2) | 10 (4.6) | 35 (16.4) | 8 (4.0) | 67 (13.7) | 20 (8.7) |

[a]Total number of eyes diagnosed with trichiasis in trachoma surveys. [b] Trichiasis diagnosis and severity from expert grading for eyes diagnosed with trichiasis in trachoma surveys.

‡Among eyes diagnosed with trichiasis in both the trachoma survey and this study (excluding those without trichiasis during the expert examination; Minor trichiasis = ≤5 eyelashes toughing the eyeball or evidence of epilation in <1/3rd of the eyelid; Major trichiasis => 5 eyelashes toughing the eyeball or evidence of epilation in ≥1/3rd of the eyelid [a] No visual acuity data for 31 eyes; [b] No visual acuity data for 6 eyes.

## Primary outcome: Negative predictive value

The summary statistics for the scarring diagnosis between the survey grading (TS, diagnosis of interest) and the expert photographic grading (TCS status), are presented for each study site in Table 5a. In Ethiopia, a total of 593 eyes had paired tarsal conjunctival scaring data from the survey and expert photo grading. Among the 111 (18.7%) eyes that were graded as having no TS in the survey, 103 (92.8%) were found to have TCS (false negative rate) in the expert photographic grading, giving a NPV of 7.2% (95% CI 3.2% - 13.7%). In Uganda, a total of 215 eyes had paired survey and expert photo grading data. Among these, 28 (24.3%) eyes were graded as having no TS in the survey grading, of which 19 (67.9%) were found to have TCS (false negative rate) in the expert photographic grading, giving a NPV of 32.1% (95% CI 15.9% - 52.4%). In Nigeria, a total of 448 eyes had paired survey and expert photo grading data. Among the 111 (24.8%) eyes that were graded as having no TS in the survey grading, 100 (90.1%) were found to have TCS (false negative rate) in the expert photographic grading, giving a NPV of 9.9% (95% CI 5.0% - 17.0%).

## Secondary Outcomes

**Tarsal conjunctival scarring grading further comparisons.** In Ethiopia, among eyes identified as having TS in the survey grading (482), only 1.7% were graded as not having TCS in the expert photo grading (false positive rate); PPV, 98.3% (95% CI 96.8% - 99.3%). The sensitivity of TS survey grading was 82.1% (95% CI, 78.8% - 85.2%), while the specificity was 50.0% (24.7% - 75.3%), Table 5a. Similar trends were seen with the survey and detailed TCS field grading comparison Table 5b.

**Table 4. Tarsal conjunctival scarring frequency in eyes identified with trichiasis in the trachoma surveys and comparisons, by study site.**

| Characteristics | | All three countries | | Ethiopia | | Uganda | | Nigeria | |
|---|---|---|---|---|---|---|---|---|---|
| | | Trichiasis case eyes N=1324 (%) | Comparison eyes N=652 (%) | Trichiasis case eyes N=594 (%) | Comparison eyes N=220 (%) | Trichiasis case eyes N=223 (%) | Comparison eyes N=200 (%) | Trichiasis case eyes N=507 (%) | Comparison eyes N=232 (%) |
| **TS – survey grading** | | | | | | | | | |
| No | | 264 (20.0) * | | 111 (18.7) * | | 31 (13.9) | | 122 (24.1) | |
| Yes | | 1058 (80.0) | | 482 (81.3) | | 192 (86.1) | | 384 (75.9) | |
| Total with data | | 1322 (100) | | 593 (100 | | 223 (100) | | 506 (100) | |
| Ungradable | | 1 | | 0 | 0 | 0 | | 1 | |
| **TCS – detailed field grading summary** | | | | | | | | | |
| No | | 116 (8.9) | 267 (41.1) | 29 (4.9) | 66 (30.1) | 24 (11.2) | 94 (47.0) | 63 (12.9) | 107 (46.3) |
| Yes | | 1181 (91.1) | 383 (58.9) | 564 (95.1) | 153 (69.9) | 191 (88.8) | 106 (53.0) | 426 (87.1) | 124 (53.7) |
| Total with data | | 1297 (100) | 650 (100) | 593 (100) | 219 (100) | 215 (100) | 200 (100) | 489 (100) | 231 (100) |
| Ungradable | | 27 | 2 | 1 | 1 | 8 | 0 | 18 | 1 |
| **TCS – detailed field grading** | | | | | | | | | |
| S0 | | 116 (8.9) | 267 (41.1) | 29 (4.9) | 66 (30.1) | 24 (11.2) | 94 (47.0) | 63 (12.9) | 107 (46.3) |
| S1 | S1a | 90 (6.9) | 118 (18.1) | 23 (3.9) | 42 (19.2) | 15 (7.0) | 31 (15.5) | 52 (10.6) | 45 (19.5) |
| | S1b | 67 (5.2) | 58 (8.9) | 39 (6.6) | 28 (12.8) | 10 (4.6) | 8 (4.0) | 18 (3.7) | 22 (9.5) |
| | S1c | 169 (13.0) | 112 (17.2) | 101 (17.0) | 54 (24.7) | 46 (21.4) | 51 (25.5) | 22 (4.5) | 7 (3.0) |
| S2 | | 309 (23.8) | 56 (8.6) | 140 (23.6) | 21 (9.6) | 84 (39.1) | 16 (8.0) | 85 (17.4) | 19 (8.2) |
| S3 | | 546 (42.1) | s | 261 (43.0) | 8 (3.6) | 36 (16.7) | 0 | 249 (50.9) | 31 (13.4) |
| Ungradable | | 27 | 2 | 1 | 1 | 8 | 0 | 18 | 1 |
| **TCS – expert photo grading summary** | | | | | | | | | |
| No | | 102 (8.1) | 327 (52.8) | 16 (2.7) | 61 (27.9) | 54 (24.7) | 167 (86.1) | 32 (7.2) | 99 (48.1) |
| Yes | | 1153 (91.9) | 292 (47.2) | 577 (97.3) | 158 (72.1) | 161 (74.9) | 27 (13.9) | 415 (92.8) | 107 (51.9) |
| Total with data | | 1255 (100) | 619 (100) | 593 (100) | 219 (100) | 215 (100) | 194 (100) | 447 (100) | 206 (100) |
| Ungradable | | 1 | 5 | 0 | 0 | 0 | 0 | 1 | |
| **TCS – expert photo grading** | | | | | | | | | |
| S0 | | 102 (8.1) | 327 (52.8) | 16 (2.7)[a] | 61 (27.9)[⊥] | 54 (25.1) [c] | 167 (86.1)[Ω] | 32 (7.2)[ψ] | 99 (48.1)[τ] |
| S1 | S1a | 31 (2.5) | 23 (3.7) | 9 (1.5) | 17 (7.8) | 8 (3.7) | 1 (0.5) | 14 (3.1) | 5 (2.4) |
| | S1b | 36 (2.9) | 44 (7.1) | 14 (2.4) | 27 (12.3) | 8 (3.7) | 3 (1.6) | 14 (3.1) | 14 (6.8) |
| | S1c | 146 (11.6) | 95 (15.2) | 60 (10.1) | 47 (21.5) | 28 (13.0) | 10 (5.1) | 58 (13.0) | 38 (18.4) |
| S2 | | 365 (29.1) | 64 (10.3) | 230 (38.8) | 41 (18.7) | 44 (20.5) | 6 (3.1) | 91 (20.4) | 17 (8.3) |
| S3 | | 575 (45.8) | 66 (10.6) | 264 (44.5) | 26 (11.9) | 73 (34.0) | 7 (3.6) | 238 (53.2) | 33 (16.0) |
| Ungradable | | 1 | 5 | 0 | 0 | 0 | 0 | 1 | 5 |

TCS = Tarsal Conjunctival Scarring; TS = Trachomatous Scarring (WHO Simplified System); *One person with trichiasis had no TS data in one eye from the survey grading; [⊥] No photo for one eye with ungradable tarsus; [Ω] No photo for 6 eyes; [ψ] No photo for 59 eyes; [c] No photo for 8 eyes with ungradable tarsus; [τ] No photo for 21 eyes; [a] No photo for one eye with ungradable tarsus.

In Uganda, among eyes identified as having TS in the survey grading (187), 24.1% were graded as not having TCS in the expert photo grading (false positive rate), giving the lowest PPV of the three study sites: 75.9% (95% CI 69.2% - 81.9%). Sensitivity was 88.2% (95% CI, 82.2% - 92.7%), while specificity was 16.7% (7.9% - 29.3%), Table 5a. The NPV value for the survey vs detailed field grading was similar to the survey vs expert photographic grading, but with lower PPV (92.0%, 95% CI 87.1% - 95.4%) and higher specificity (37.5%, 95% CI 18.8% - 59.4%), Table 5b.

**Table 5. Tarsal conjunctival grading in eyes diagnosed with trichiasis by study site.**

**a) Survey vs expert photo grading**

| Expert photo grading (reference) | All three countries | | | Ethiopia | | | Uganda | | | Nigeria | | |
|---|---|---|---|---|---|---|---|---|---|---|---|---|
| | Survey grading | | | Survey grading | | | Survey grading | | | Survey grading | | |
| | No TS | Yes TS | Total | No TS | Yes TS | Total | No TS | Yes TS | Total | No TS | Yes TS | Total |
| No TCS (%) | 28 (11.2) | 74 (7.4) | 102 (8.1) | 8 (7.2) | 8 (1.7) | 16 (2.7) | 9 (32.1) | 45 (24.1) | 54 (25.0) | 11 (9.9) | 21 (6.2) | 32 (7.1) |
| Yes TCS (%) | 222 (88.8) | 932 (92.6) | 1154 (91.9) | 103 (92.8) | 474 (98.3) | 577 (97.3) | 19 (67.9) | 142 (75.9) | 161 (74.9) | 100 (90.1) | 316 (93.8) | 416 (92.9) |
| Total (%) | 250 (100) | 1006 (100) | 1256 (100) | 111 (100) | 482 (100) | 593 (100) | 28 (100) | 187 (100) | 215 (100) * | 111 (100) | 337 (100) | 448 (100) * |
| Sensitivity (%) | 80.8 | (78.4 – 83.0) | | 82.1 | (78.8 – 85.2) | | 88.2 | 82.2 – 92.7 | | 76.0 | 71.6 – 80.0 | |
| Specificity (%) | 27.5 | (19.1 – 37.2) | | 50.0 | (24.7 – 75.3) | | 16.7 | 7.9 – 29.3 | | 34.4 | 18.6 – 53.2 | |
| PPV (%) | 92.6 | (90.9 – 94.2) | | 98.3 | (96.8 – 99.3) | | 75.9 | 69.2 – 81.9 | | 93.8 | 90.6 – 96.1 | |
| NPV (%) | 11.2 | (7.6 – 15.8) | | 7.2 | (3.2 – 13.7) | | 32.1 | 15.9 – 52.4 | | 9.9 | 5.0 – 17.0 | |

**b) Survey vs detailed field grading**

| Detailed field grading (reference) | Survey grading | | | Survey grading | | | Survey grading | | | Survey grading | | |
|---|---|---|---|---|---|---|---|---|---|---|---|---|
| | No TS | Yes TS | Total | No TS | Yes TS | Total | No TS | Yes TS | Total | No TS | Yes TS | Total |
| No TCS (%) | 36 (14.0) | 80 (7.7) | 116 (8.9) | 12 (10.8) | 17 (3.5) | 29 (4.5) | 9 (32.1) | 15 (8.0) | 24 (11.2) | 15 (12.6) | 48 (12.9) | 119 (100) |
| Yes TCS (%) | 222 (86.0) | 960 (92.3) | 1182 (91.1) | 99 (89.2) | 465 (96.5) | 564 (95.1) | 19 (67.7) | 172 (92.0) | 191 (88.8) | 104 (87.4) | 323 (87.1) | 371 (100) |
| Total (%) | 258 (100) | 1040 (100) | 1298 (100) | 111 (100) | 482 (100) | 593 (100)a | 28 (100) | 187 (100) | 215 (100) | 119 (100) | 371 (100) | 490 (100)‡ |
| Sensitivity (%) | 81.2 | (78.9 – 83.4) | | 82.4 | (79.1 – 85.5) | | 90.1 | (84.9 – 93.9) | | 87.1 | (83.2 – 90.3) | |
| Specificity (%) | 31.0 | (22.8 – 40.3) | | 41.4 | (23.5 – 61.1) | | 37.5 | (18.8 – 59.4) | | 12.6 | (7.2 – 19.9) | |
| PPV (%) | 92.3 | (90.5 – 93.9) | | 96.5 | (94.4 – 97.9) | | 92.0 | (87.1 – 95.4) | | 75.6 | (71.3 – 79.6) | |
| NPV (%) | 14.0 | (10.0 – 18.8) | | 10.8 | (5.7 – 18.1) | | 32.1 | (15.9 – 52.4) | | 23.8 | (14.0 – 36.2) | |

*Among eyes with gradable photographs

aNo TS data for one eye in the survey due to inability to evert the eyelid.

‡No exam data for 8 eyes due to inability to ever the eyelid

TCS = Tarsal Conjunctival Scarring

TS = Trachomatous Scarring (WHO Simplified System)

In Nigeria, among eyes identified as having TS in the survey grading (337), 6.2% were graded as not having TCS in the expert photo grading (false positive rate), PPV 93.8% (95% CI 90.6% - 96.1%). Sensitivity was 76.0% (95% CI, 71.6% - 80.0%) and specificity was 34.4% (18.6% - 53.2%), Table 5a. The survey vs detailed field grading provided slightly higher values than the survey vs photographic grading in NPV and sensitivity, Table 5b.

**Misdiagnosed TS.** In total the majority of eyes (174/250, 69.6%) that were diagnosed as not having TS in the survey grading but diagnosed as having TCS in all the three settings were those with extensive tarsal conjunctival scarring (patches of scarring occupying ≥1/3 of the upper tarsal conjunctiva, grade S2 and S3) in the expert photographic grading. Table 6a, Fig 2a. On the other hand, there was comparable grading between the photo grading and the detailed field grading with only 8.5% of eyes with extensive tarsal conjunctival scarring in the expert photographic grading were diagnosed as not having TCS by the detailed field grading Table 6b.

**Table 6. TS diagnoses in the survey and detailed field grading compared to the photographic grading, by study site.**

a)TS survey grading

| | | All three countries | | Ethiopia N = 593* | | Uganda N = 215* | | Nigeria n = 448* | |
|---|---|---|---|---|---|---|---|---|---|
| | | No TS n/250 (%) | Yes TS n/1006 (%) | No TS n/111 (%) | Yes TS n/482 (%) | No TS n/28 (%) | Yes TS n/187 (%) | No TS n/111 (%) | Yes TS n/337 (%) |
| Photo grading (reference) | | | | | | | | | |
| S0 | | 28 (11.2) | 74 (7.4) | 8 (7.2) | 8 (1.7) | 9 (32.1) | 45 (24.1) | 11 (9.9) | 21 (6.2) |
| S1 | S1a | 7 (2.8) | 24 (2.4) | 2 (1.8) | 7 (1.4) | 2 (7.1) | 6 (3.2) | 3 (2.7) | 11 (3.3) |
| | S1b | 14 (5.6) | 22 (2.2) | 7 (6.3) | 7 (1.4) | 1 (3.6) | 7 (3.7) | 6 (5.4) | 8 (2.4) |
| | S1c | 27 (10.8) | 119 (11.8) | 9 (8.1) | 51 (10.6) | 3 (10.7) | 25 (13.4) | 15 (13.5) | 43 (12.8) |
| S2 | | 63 (52.2) | 302 (30.0) | 43 (38.7) | 187 (38.8) | 5 (17.9) | 39 (20.9) | 15 (13.5) | 76 (22.6) |
| S3 | | 111 (44.4) | 465 (46.2) | 42 (37.8) | 222 (46.1) | 8 (28.6) | 65 (34.8) | 61 (54.9) | 178 (52.8) |

b)TCS detailed field grading

| | | No TCS n/129 (%) | Yes TCS n/1333 (%) | No TCS n/47 (%) | Yes TCS n/761 (%) | No TCS n/24 (%) | Yes TCS n/188 (%) | No TCS n/58 (%) | Yes TCS n/384 (%) |
|---|---|---|---|---|---|---|---|---|---|
| Photo grading (reference) | | | | | | | | | |
| S0 | | 65 (50.4) | 47 (3.5) | 20 (42.6) | 7 (0.9) | 21 (87.5) | 32 (17.0) | 24 (41.4) | 8 (2.1) |
| S1 | S1a | 17 (13.2) | 18 (1.3) | 6 (12.8) | 7 (0.9) | 1 (4.2) | 7 (3.7) | 10 (17.2) | 4 (1.0) |
| | S1b | 12 (9.3) | 33 (2.5) | 3 (6.4) | 21 (2.8) | 1 (4.2) | 7 (3.7) | 8 (13.8) | 5 (1.3) |
| | S1c | 24 (18.6) | 161 (12.1) | 12 (25.5) | 87 (11.4) | 1 (4.2) | 27 (14.4) | 11 (19.0) | 47 (12.2) |
| S2 | | 8 (6.2) | 440 (33.0) | 4 (8.5) | 309 (40.6) | 0 (0.0 | 44 (23.4) | 4 (6.9) | 87 (22.7) |
| S3 | | 3 (2.3) | 634 (47.6) | 2 (4.3) | 330 (43.4) | 0 (0.0 | 71 (37.8) | 1 (1.7) | 233 (60.7) |

*Among eyes with gradable photographs.

TCS = Tarsal Conjunctival Scarring.

TS = Trachomatous Scarring (WHO Simplified System).

**Characteristics of eyes with and without TCS.** Eyes with any TCS as assessed by expert photo grading had more severe entropion and trichiasis, more entropic, metaplastic, and corneal trichiatic lashes, more tarsal conjunctival inflammation (p < 0.0001 for all), and central corneal opacity (p = 0.0001), than those with no TCS, Table 7.

**Factors associated with TCS severity.** Multivariable analysis was conducted to determine conjunctival scarring severity (graded by expert from photo using the FPC grading system) association with demographic and clinical characteristics, Table 8. Older people had more severe TCS than their younger counterparts (p = 0.0016). Eyes with more severe TCS were more likely to have major trichiasis (>5 eyelashes touching the eyeball or evidence of epilation in ≥1/3rd of the eyelid) (p = 0.0395), corneal opacity (p < 0.0001), and tarsal conjunctival inflammation (p = 0.0109) than those with no or milder grade of TCS. Eyes with more severe TCS were also more likely to have corneal than peripheral trichiatic eyelashes (p = 0.0021).

## Discussion

The key findings from this study were that field grading of TS missed a significant proportion of eyes with TCS as graded by experts using a detailed grading system, including those with extensive scarring, across all three countries, giving a high false negative rate. This might have occurred for several reasons. Survey graders are non-expert trained field personnel which were only trained to diagnose TS using slides. There was no specific standardisation or certification process for the accurate grading of TS, unlike for TF grading. The WHO simplified trachoma grading system definition of TS is "…easily visible as white lines, bands or sheets… [7,15,21]. However, conjunctival scarring may be mild or obscured by

**(a) Diagnosed as "no TS" in trachoma surveys**    **(b) Diagnosed as "TS" in trachoma surveys**

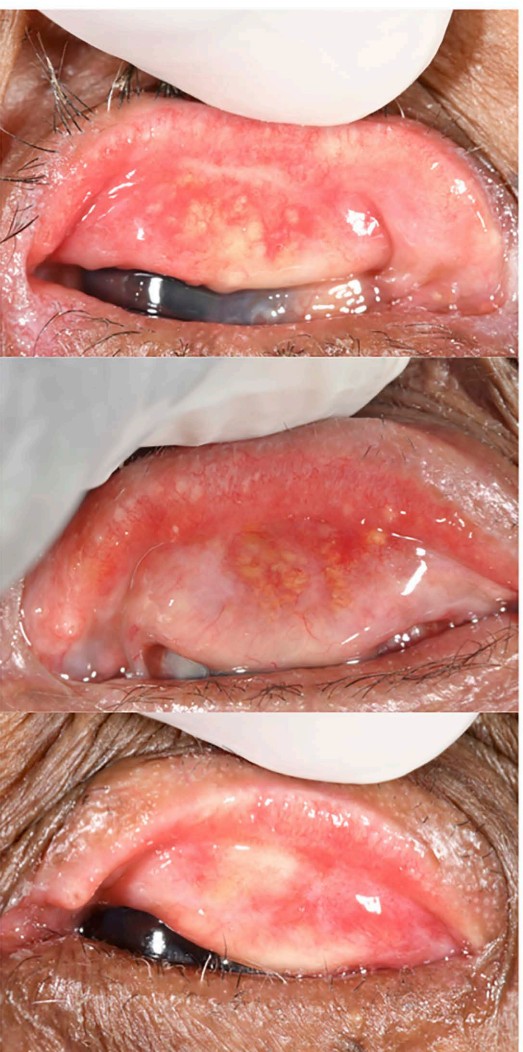 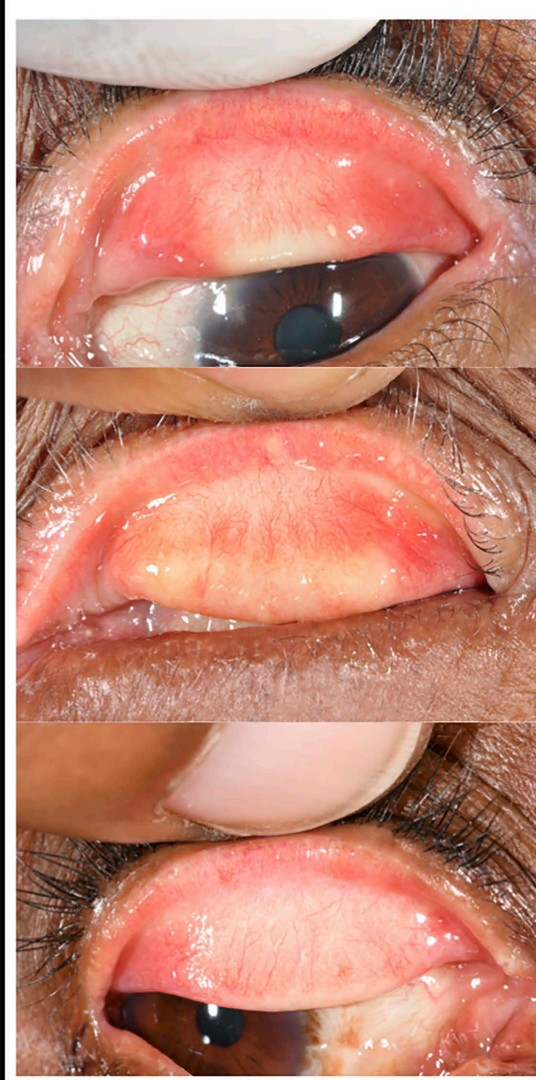

Note: the trachoma survey graders have misclassified the eyes with advanced tarsal conjunctival scarring as "no TS" (the eyes under "(a)"); while they have correctly graded the ones with milder tarsal conjunctival scarring (the eyes under "(b)").

**Fig 2. Illustration of eyes diagnosed as having no TS and with TS in trachoma surveys.**

conjunctival inflammation, which is quite common in people with trichiasis and as shown in this study data. Studies using finer grading systems of conjunctival scarring in trachoma-endemic populations showed that conjunctival scarring is a very broad clinical phenotype, ranging from obvious and highly distorting, down to almost microscopic [14,22]. This may result in missing the diagnosis of milder scarring in a survey setting. Another possible reason is the likely discrepancy between field and photographic grading. A study in Ethiopia showed that agreements between field and photographic grading for TS are lower than agreements for TF, suggesting likely diagnostic errors in field settings [23]. However, even taking these factors into account, there was still a significant mismatch between the survey grading and the repeat detailed expert field grading across the three study settings.

**Table 7. Characteristics of eyes with and without tarsal conjunctival scarring in people with at least one trichiatic eye.**

| Characteristics | Eyes without TCS n/113 (%) | Eyes with TCS n/1361 (%) | OR | 95% CI | P value |
|---|---|---|---|---|---|
| **Trichiasis** | | | | | |
| Eyes with no trichiasis or epilation | 78 (69.0) | 457 (33.6) | 1.93 | (1.53 – 2.43) | <0.0001 |
| Epilated successfully (diagnosed by epilation evidence) | 5 (4.4) | 62 (4.6) | | | |
| 1–5 eyelashes | 20 (17.7) | 591 (43.4) | | | |
| 6+eyelashes | 10 (8.8) | 251 (18.4) | | | |
| **Trichiasis Severity‡** (incudes epilation) | | | | | |
| No eyelashes | 78 (69.0) | 457 (33.6) | 2.60 | (1.72 – 3.94) | <0.0001 |
| Minor | 21 (18.6) | 552 (40.6) | | | |
| Major | 14 (12.4) | 352 (25.9) | | | |
| **Entropion** | | | | | |
| None | 68 (60.2) | 237 (17.4) | 2.43 | (1.73 – 3.41) | <0.0001 |
| Mild | 19 (16.8) | 382 (28.1) | | | |
| Moderate | 15 (13.3) | 517 (38.0) | | | |
| Severe | 10 (8.8) | 163 (12.0) | | | |
| Complete | 1 (0.9) | 62 (4.6) | | | |
| **Eyelash Type‡** | | | | | |
| No trichiasis | 78 (69.0) | 457 (33.6) | 1 | | |
| Epilated successfully | 5 (4.4) | 62 (4.6) | 2.09 | (0.70 – 6.29) | 0.1878 |
| Entropic | 10 (8.8) | 286 (21.0) | 4.83 | (2.29 – 10.17) | <0.0001 |
| Metaplastic | 12 (10.6) | 378 (27.8) | 5.32 | (2.71 – 10.43) | <0.0001 |
| Misdirected | 6 (5.3) | 53 (3.9) | 1.49 | (0.62 – 3.60) | 0.3737 |
| Mixed | 3 (2.6) | 126 (9.3) | 7.09 | (2.17 – 23.18) | 0.0012 |
| **Eyelash Location‡** | | | | | |
| No trichiasis | 78 (69.0) | 457 (33.6) | 1 | | |
| Epilated successfully | 5 (4.4) | 62 (4.6) | 2.12 | (0.70 – 6.36) | 0.1815 |
| Corneal only | 6 (5.3) | 475 (34.9) | 13.51 | (5.12 – 35.68) | <0.0001 |
| Peripheral only | 15 (13.3) | 112 (8.2) | 1.27 | (0.70 – 2.30) | 0.4219 |
| Mixed location | 9 (8.0) | 255 (18.7) | 4.84 | (2.19 – 10.66) | 0.0001 |
| **Tarsal conjunctival inflammation [a]** | | | | | |
| None | 77 (68.1) | 348 (25.6) | 3.18 | (2.35 – 4.31) | <0.0001 |
| Mild | 26 (23.0) | 269 (19.8) | | | |
| Moderate | 7 (6.2) | 421 (31.2) | | | |
| Severe | 2 (1.8) | 313 (23.2) | | | |
| **Lower eyelid trichiasis** | | | | | |
| No | 111 (98.2) | 1291 (94.9) | 3.01 | (0.74 – 12.31) | 0.1253 |
| Yes | 2 (1.8) | 70 (5.1) | | | |
| **Corneal Opacity [b]** | | | | | |
| None (CC0) | 76 (67.3) | 562 (42.0) | 1.51 | (1.23 – 1.84) | 0.0001 |
| Peripheral (CC1) | 10 (8.8) | 185 (13.8) | | | |
| Off Central Opacity (CC2a, C2b) | 9 (8.0) | 149 (11.1) | | | |
| Central Opacity (CC2c CC2d, CC3) | 18 (15.9) | 442 (33.0) | | | |

Analyses done among all eyes of people diagnosed with trichiasis at least in one eye, adjusted for between eye correlation. TCS grading is based on expert photographic grading.

‡Among eyes diagnosed with trichiasis, Minor trichiasis = ≤5 eyelashes toughing the eyeball or evidence of epilation in <1/3rd of the eyelid; Major trichiasis =>5 eyelashes toughing the eyeball or evidence of epilation in ≥1/3rd of the eyelid

[a]ungradable one eye, [b] 23 Phthisic eyes excluded

**Table 8. Factors associated with Tarsal conjunctival scarring severity in multivariable analyses.**

| | TCS photographic grading using the FPC system | | | | Univariable analysis | | | Multivariable analysis | | |
|---|---|---|---|---|---|---|---|---|---|---|
| | No n/440 (%) | Mild n/199 (%) | Moderate n/1079 (%) | Severe n/375 (%) | OR | 95% CI | P value | OR | 95% CI | P value |
| **Characteristics** | | | | | | | | | | |
| **Age, mean (SD)** | 48.6 (16.4) | 54.6 (17.2) | 54.6 (17.0) | 60.2 (16.6) | 1.02 | 1.01 – 1.03 | 0.0002 | 1.016 | 1.006 – 1.03 | 0.0016 |
| **Sex** | | | | | | | | | | |
| Male | 167 (27.9) | 56 (9.4) | 277 (46.3) | 98 (16.4) | 1.17 | 0.81 – 1.70 | 0.3928 | 1.25 | 0.84 – 1.87 | 0.2668 |
| Female | 273 (18.0) | 143(9.5) | 802 (53.6) | 277 (18.5) | | | | | | |
| **Entropion** | | | | | | | | | | |
| None | 320 (44.4) | 112 (15.5) | 249 (34.5) | 40 (5.6) | 1.50 | 1.27– 1.78 | <0.0001 | 1.18 | 0.99 – 1.42 | 0.0642 |
| Mild | 82 (14.6) | 53 (9.4) | 328 (58.4) | 99 (17.6) | | | | | | |
| Moderate | 27 (4.7) | 30 (5.2) | 380 (66.2) | 137 (23.9) | | | | | | |
| Severe | 10 (5.8) | 4 (2.3) | 99 (57.2) | 60 (34.7) | | | | | | |
| Complete | 1 (1.6) | 0 | 23 (36.5) | 39 (61.9) | | | | | | |
| **Trichiasis severity‡** (*incudes epilation*) | | | | | | | | | | |
| No eyelashes | 78 (14.6) | 75 (14.0) | 328 (61.3) | 54 (10.1) | 2.35 | 1.73 – 3.20 | <0.0001 | 1.47 | 1.02 – 2.12 | 0.0395 |
| Minor | 21 (3.7) | 34 (5.9) | 390 (68.1) | 128 (22.3) | | | | | | |
| Major | 14 (3.8) | 8 (2.2) | 191 (52.2) | 153 (41.8) | | | | | | |
| **Eyelash location‡** | | | | | | | | | | |
| No trichiasis | 78 (14.6) | 75 (14.0) | 328 (61.3) | 54 (10.1) | | | | | | |
| Epilated successfully | 5 (7.5) | 0 | 38 (56.7) | 24 (35.8) | 3.19 | 1.48 – 6.89 | 0.0031 | 2.29 | 1.01 – 5.24 | 0.0145 |
| Corneal only | 6 (1.2) | 21 (4.4) | 310 (64.4) | 144 (29.9) | 2.80 | 1.67 – 4.69 | 0.0001 | 2.17 | 1.33 – 3.56 | 0.0021 |
| Peripheral only | 15 (11.8) | 12 (9.4) | 76 (59.8) | 24 (18.9) | 1 | | | 1 | | |
| Mixed location | 9 (3.4) | 9 (3.4) | 157 (59.5) | 89 (33.7) | 3.11 | 1.82 – 5.33 | <0.0001 | 1.83 | 1.08 – 3.10 | 0.0236 |
| ***Corneal opacity*** [b] | | | | | | | | | | |
| None (CC0) | 379 (32.7) | 152 (13.1) | 543 (46.8) | 85 (7.3) | 1.64 | 1.46 – 1.85 | <0.0001 | 1.44 | 1.26 – 1.64 | <0.0001 |
| Peripheral (CC1) | 20 (8.8) | 16 (7.1) | 139 (61.5) | 51 (22.6) | | | | | | |
| Off Central Opacity (CC2a, C2b) | 16 (8.7) | 11 (6.0) | 110 (60.1) | 46 (25.1) | | | | | | |
| Central Opacity (CC2c CC2d, CC3) | 25 (5.0) | 18 (3.4) | 280 (55.8) | 179 (53.7) | | | | | | |
| **Tarsal conjunctival inflammation** [a] | | | | | | | | | | |
| None | 325 (39.4) | 109 (13.2) | 322 (39.1) | 68 (8.2) | 1.44 | 1.25 – 1.65 | <0.0001 | 1.21 | 1.04 – 1.39 | 0.0109 |
| Mild | 101 (23.0) | 64 (14.6) | 220 (50.1) | 54 (12.3) | | | | | | |
| Moderate | 11 (2.3) | 23 (4.8) | 315 (66.3) | 126 (26.5) | | | | | | |
| Severe | 2 (0.6) | 2 (0.6) | 222 (64.5) | 118 (34.3) | | | | | | |

Analysis is adjusted for between eye correlation.

‡Among eyes diagnosed with trichiasis, Minor trichiasis = ≤5 eyelashes toughing the eyeball or evidence of epilation in <1/3rd of the eyelid; Major trichiasis = >5 eyelashes toughing the eyeball or evidence of epilation in ≥1/3rd of the eyelid.

[a]ungradable one eye

[b]23 Phthisic eyes excluded

We have explored the assumption that the type of scarring being missed in the survey grading is probably mild as it would not meet the WHO definition of "…easily visible…". For instance, about half of the scarring that was missed in the expert field grading compared to the expert photographic grading was scarring occupying <1/3 of the upper tarsal conjunctiva. Mild scarring can be difficult to grade when the field examiner only uses a 2.5×magnifying loupe. It may only be visible with higher magnification or involve deeper tissue around eyelash follicles. Moreover, in the

presence of trichiasis, the tarsal conjunctiva is often inflamed, which can obscure the underlying scarring. In addition, when a phenotype is subtle, there is likely to be more inter-observer variation and less reliable grading. However, this was not the case in the survey grading. Among the eyes with miscategorised scarring in the survey, about three-quarters in Ethiopia and Nigeria, and nearly half in Uganda, were those with extensive patches of scarring occupying ≥1/3 of the upper tarsal conjunctiva, which are likely to be easily visible with a loupe magnification. In addition, specificity was very low, suggesting that the survey graders were calling someone without TS as having TS. These results clearly show that diagnosing TS in a survey setting by non-expert field graders is difficult, regardless of the severity of the scarring.

TCS was common, not only in trichiasis cases but also in people without trichiasis, particularly in Ethiopia and Nigeria. These results are consistent with previous studies. In a case-control study conducted in Amhara, Ethiopia, almost all major trichiasis cases and 73.1% of the eyes of non-trichiatic controls had some degree of TCS [18]. In this hyperendemic setting study, it was a challenge to recruit controls without TCS as most of the adults triaged were found to have some degree of TCS [18]. A trial that enrolled minor trichiasis cases from a trachoma hyperendemic area in Amhara, Ethiopia, reported that about 99% of trichiasis cases had clinically apparent TCS [24]. However, unlike in the programmatic field surveys, the examinations for the study participants for the above two Amhara studies [18,24] were done by expert ophthalmologists. It is likely that the reported proportion of trichiasis cases identified with TS would be lower in population-based surveys (where graders have less expertise). Population-based surveys conducted in 152 districts of Amhara region between 2010 and 2015 showed 23.3% of the trichiasis cases had concurrent TS [25].

In addition, areas with a low burden of endemic trachoma might have proportionally more trichiasis cases without easily visible TCS than those with relatively higher burden. This is probably the reason why the scarring frequency in trichiatic eyes in Uganda is lower than the other two study sites. This may indicate less severe disease in Uganda as it is closer to achieving elimination compared to Nigeria and Ethiopia [16].

Trichiasis can be caused by ocular pathologies other than trachoma, such as blepharitis, Steven Johnson syndrome, burns, trauma, herpes zoster and ocular cicatricial pemphigoid [26]. However, these well-recognised differential diagnoses are relatively uncommon in trachoma-endemic settings and often involve a degree of TCS as well. However, it is unknown, in eyes with trichiasis, how much and what type of scarring is pathologically relevant to define it as trachomatous.

In Uganda, the frequency of TCS in the comparison participants was lower, at 14%, than in Ethiopia and Nigeria. This, however, was not unanticipated as the comparisons in Uganda were, younger, educated, with jobs, married, and male dominated than the trichiasis cases. These parameters often predict risk and disease severity in trachoma [26]. A similar trend was seen in the Nigeria comparisons in terms of age and educational status. Old age was one of the independent predictors of severe TCS in our study, consistent with findings elsewhere, such as a population-based study in Amhara where TS was significantly associated with older age [25].

We have explored the characteristics of eyes with and without TCS from the photo grading. We have found that TCS was significantly associated with signs of severe disease for trachoma. These results resonate very well with the assumption that TCS is key indicator of an advanced stage of disease in trachoma, and the more severe the scarring the more severe the disease.

This is the first population-based multi-country study that explored the magnitude of TCS in three different settings of varying trachoma burden. It however has several limitations. Finding the cases that were examined by the population-based survey team was arduous as there was no complete address data that could help identify the correct person. The field work team primarily relied on the household head name collected during the trachoma surveys and then confirmation was provided by the local administrators and community health personnel to find the correct person for the repeat examination. All the teams in the three countries were provided with GPS devices to help them locate the households of the people examined in the trachoma surveys using geolocation data collected during the surveys. This was employed adequately in the Ethiopia field work, particularly to locate the households of survey participants whose

evaluation unit was recorded incorrectly after it has been changed due to security related reasons. However, this GPS tracking system was not uniformly used in the Uganda and Nigeria data collection, limiting the ability of the team to locate the correct person. Another limitation is the expert field grading in Nigeria was conducted after six months of the initial survey due to security related challenges. It is highly unlikely that the TCS progressed within 12 months' time, but still there could be a change between a scar being easily visible or not, particularly in relation to a change in tarsal conjunctival inflammation. Field and photographic grading are likely to yield some degree of difference particularly in identifying milder cases regardless of the expertise of the field grader. We have observed this in this study. This conforms with the assumption that field grading of conjunctival scarring is challenging and susceptible to error. The TT definition for surveys conducted prior to 2019 was not limited to upper eyelid. However, this would not affect the results of this study as the three countries where this study is conducted in practice limited the definition of TT to the upper eyelid only. Considerable numbers of trichiasis cases were not found in both Uganda and Nigeria for the expert field grading. Although the "found" were not significantly different from the "not found" in terms of age and sex, it is still possible that these two groups of trichiasis cases may have different clinical features that possibly may influence the result of this study. There is no complete data on the triaging process of comparisons from Uganda and Nigeria for eligibility. As a multi-country study, the findings are likely to be generalizable beyond a single geographic, cultural, or health system context. However, it remains important to interpret the results considering local variations, as country-specific confounding factors may affect internal validity or obscure true associations. For instance, in Uganda, the number of trichiatic eyes without TS was relatively small, which reduces the precision of the NPV estimates and thereby its generalisability.

Overall, the data from the three study settings of varying burden of trachoma showed that, including TS to define a trichiasis "trachomatous" could result in underestimating the burden of TT. This risks the sights of hundreds of thousands of affected individuals who would be excluded from management planning within trachoma elimination programmes worldwide. The varying phenotype of TCS, from subtle to distorting, limit the use of TS as one of the indicators to inform trachoma elimination. Furthermore, the subjectivity of "easily visible" and lack of clarity on the extent of scarring that warrant trichiasis to be trachomatous is not clear. What is visible for one person might not be visible to another just from personal differences that could be influenced by several factors. The data from this study also strongly suggest that the presence of TCS and its severity can be effectively used to determine the severity of trichiasis and decide the management type. We do not recommend that programs collect TS data in the field to determine if trichiasis is trachomatous or not. However, should a program want to collect and use TCS data to inform disease severity, they could consider the use of machine learning to diagnose TCS from photographs.

## Supporting information

**S1 Fig. Everted tarsus showing the gradable area for tarsal conjunctival scarring.** (TIF)

**S1 Checklist. The area within the dotted lines is used to assess tarsal conjunctival scarring, following WHO guidelines, using 2.5× binocular loupes and adequate lighting (sunlight or torch).** (DOCX)

## Acknowledgments

We are extremely grateful for the generous funding and project oversight support that we received from Sightsavers International. We would like to express our deepest gratitude to the persons with trichiasis, community members, and data collectors that participated in this study in Ethiopia, Nigeria and Uganda. We are also hugely indebted to the Ministry of Health of Ethiopia, Nigeria and Uganda, and their respective regional or state health bureaus, zonal, district and local government offices for providing the data and coordinating the data collection. We are extremely grateful to

Tropical Data team, The Carter Center, RTI International, and Sightsavers that supported the project and the data collection as part of their trachoma elimination programme in the three study countries. We are also hugely indebted to the project and financial coordination we received from the project support staff at the International Centre for Eye Health, LSHTM.

## Author contributions

**Conceptualization:** Esmael Habtamu, Scott Mcpherson, Emma M. Harding-Esch, David Macleod, Victor Hu, Paul Courtright, Matthew Burton.

**Data curation:** Esmael Habtamu, Caleb Mpyet, Francis Mugume, Gladys Atto, Gilbert Baayenda, Ana Bakhtiari.

**Formal analysis:** Esmael Habtamu, Fikre Seife, David Macleod.

**Funding acquisition:** Esmael Habtamu, Tom Millar, Paul Courtright, Matthew Burton.

**Investigation:** Esmael Habtamu, Caleb Mpyet, Gladys Atto.

**Methodology:** Esmael Habtamu, Ana Bakhtiari, Emma M. Harding-Esch, David Macleod, Victor Hu, Paul Courtright, Matthew Burton.

**Project administration:** Esmael Habtamu, Caleb Mpyet, Francis Mugume, Gilbert Baayenda, Sharone Backers, Zerihun Tadesse, Scott D. Nash, E. Kelly Callahan, Scott Mcpherson, Michaela Kelly.

**Resources:** Victor Hu.

**Supervision:** Esmael Habtamu, Caleb Mpyet, Francis Mugume, Sharone Backers, Zerihun Tadesse, Scott D. Nash, E. Kelly Callahan, Scott Mcpherson, Emma M. Harding-Esch, David Macleod, Michaela Kelly, Tom Millar, Victor Hu, Paul Courtright, Matthew Burton.

**Validation:** Esmael Habtamu, Victor Hu.

**Writing – original draft:** Esmael Habtamu.

**Writing – review & editing:** Caleb Mpyet, Francis Mugume, Gladys Atto, Fikre Seife, Gilbert Baayenda, Sharone Backers, Ana Bakhtiari, Zerihun Tadesse, Scott D. Nash, E. Kelly Callahan, Scott Mcpherson, Emma M. Harding-Esch, David Macleod, Michaela Kelly, Tom Millar, Victor Hu, Paul Courtright, Matthew Burton.

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
