## [Decision Letter · Decision Letter 0]

7 Apr 2025

PGPH-D-25-00302

Trichiasis With and Without Tarsal Conjunctival Scarring: a multi-country observational study

Dear Dr. Habtamu,

Thank you for submitting your manuscript to PLOS Global Public Health. After careful consideration, we feel that it has merit but does not fully meet PLOS Global Public Health’s publication criteria as it currently stands. Therefore, we invite you to submit a revised version of the manuscript that addresses the points raised during the review process.

The manuscript has been assessed by two reviewers who have provided their assessments below. They have provided some recommendations for improving the manuscript. Please carefully review their comments and make the appropriate revisions to address their concerns. 

We look forward to receiving your revised manuscript.

Kind regards,

Emma Campbell, Ph.D

Staff Editor

Journal Requirements:

2. We ask that a manuscript source file is provided at Revision. Please upload your manuscript file as a .doc, .docx, .rtf or .tex.

3. We do not publish any copyright or trademark symbols that usually accompany proprietary names, eg (R), (C), or TM  (e.g. next to drug or reagent names). Please remove all instances of trademark/copyright symbols throughout the text, including ® on page 1.

4.In the online submission form, you indicated that Data is available to any researcher under reasonable request. To facilitate the data access process please contact ethics@lshtm.ac.uk. 

3. Uploaded as supplementary information.

Reviewers' comments:

Reviewer's Responses to Questions

**Comments to the Author**

1. Does this manuscript meet PLOS Global Public Health’s publication criteria ? Is the manuscript technically sound, and do the data support the conclusions? The manuscript must describe methodologically and ethically rigorous research with conclusions that are appropriately drawn based on the data presented.

Reviewer #1: Yes

Reviewer #2: Yes

2. Has the statistical analysis been performed appropriately and rigorously?

Reviewer #1: Yes

Reviewer #2: Yes

3. Have the authors made all data underlying the findings in their manuscript fully available (please refer to the Data Availability Statement at the start of the manuscript PDF file)?

Reviewer #1: Yes

Reviewer #2: Yes

4. Is the manuscript presented in an intelligible fashion and written in standard English?

Reviewer #1: Yes

Reviewer #2: Yes

5. Review Comments to the Author

Reviewer #1: **Reviewer Comments:**

1. **Relevance and Importance of the Study:**

This study addresses a crucial question regarding the definition of trachomatous trichiasis (TT) for the purposes of determining trachoma elimination as a public health problem. The research's aim to evaluate the role of trachomatous scarring (TS) in defining TT in trachoma surveys is highly relevant and has the potential to impact how TT burden is assessed globally. By investigating the relationship between TS and Tarsal Conjunctival Scarring (TCS) grading, this study provides valuable insights into improving diagnostic accuracy and the effectiveness of intervention strategies in endemic regions.

2. **Study Design and Methodology:**

The comparative cross-sectional design is well-suited to answer the study's research questions. The multi-country approach (Ethiopia, Uganda, Nigeria) allows for diversity in the sample population and the inclusion of settings with varying levels of TT burden, which adds robustness to the findings. Furthermore, the use of expert grading to compare against field survey results strengthens the study's reliability and provides a more nuanced understanding of the relationship between TS and TT.

One potential limitation in the study design, however, is the reliance on photographic grading by independent experts. While expert grading provides a high level of accuracy, there may be some subjectivity and variability in how individual graders interpret the TCS severity. Clarification regarding how expert grading consistency was ensured (e.g., inter-rater reliability checks) would strengthen the interpretation of the findings.

3. **Data Analysis and Results:**

The study’s findings demonstrate important discrepancies between field survey grading and expert photographic grading, which shows the misclassification of trichiatic eyes with TCS. This has significant implications for public health interventions. The reported negative predictive values (NPV) and the proportion of misdiagnosed cases (particularly those with extensive TCS) highlight the risk of underestimating the burden of TT if the survey definition excludes TS. These results are presented clearly and support the study's conclusion that incorporating TCS severity could improve TT burden assessment.

However, more context on the statistical methods used to analyze the data (e.g., how the matching process was handled, any adjustments for potential confounding variables) would enhance the clarity of the results and strengthen the validity of the conclusions.

4. **Interpretation and Implications:**

The study provides compelling evidence that excluding TS from the definition of TT in surveys may result in an underestimation of TT prevalence. This has important implications for the planning and allocation of resources for trachoma elimination programs. The findings suggest that incorporating TCS severity could be a more effective way to assess TT severity and guide appropriate management strategies. However, further research is needed to determine how this new definition might impact programmatic decisions and whether the incorporation of TCS is feasible in resource-limited settings where access to expert grading might be limited.

5. **Clarity and Structure:**

The manuscript is generally well-written and structured. The background provides adequate context for understanding the significance of the study, and the methods section is detailed and easy to follow. The results are clearly presented with relevant data, and the conclusions effectively summarize the study’s findings and implications. One area for improvement could be in the discussion section, where a more thorough exploration of the limitations (such as the potential for diagnostic errors in field settings or the feasibility of incorporating TCS in large-scale surveys) would be beneficial.

6. **Suggestions for Improvement:**

- Provide more detail on the grading consistency between expert graders to address potential variability.

- Consider additional statistical analyses to account for confounding factors or explore the impact of demographic variables on the findings.

- Expand the discussion on the practical implications of incorporating TCS in field surveys, especially in terms of the feasibility in low-resource settings.

- A clearer explanation of how TCS grading can be integrated into routine survey methodologies could further clarify the study’s impact on public health practices.

7. **Conclusion:**

Overall, this study provides critical insights into the diagnostic challenges of trachoma surveys and offers a strong argument for the inclusion of TS in the definition of TT. The results are impactful and suggest significant improvements for public health interventions targeting trachoma elimination. While the study is robust, further discussion of practical challenges in implementing the findings could strengthen the manuscript.

Reviewer #2: The following point needs to be review:

1. Abstract should be little bit short

2.Review the introduction and background section to understand the context and significance of the study

3.assess the sampling strategy and data methods to determine if they are appropriate for the research question

4 evaluate the data analysis methods , result and conclusion drawn from the data . check for any errors or inconsistencies

5 the section related to trichiasis and tarsal conjunctival scarring to ensure that the definition , classification and assessment methods are clear and consistent

6. Assess the strength and limitation of the multi country observational study design. evaluate the generalizability of the findings across different countries and population

6. PLOS authors have the option to publish the peer review history of their article (what does this mean? ). If published, this will include your full peer review and any attached files.

**Do you want your identity to be public for this peer review?** For information about this choice, including consent withdrawal, please see our Privacy Policy .

Reviewer #1: **Yes: ** Dr. Ragni Kumari

Reviewer #2: No

---

## [Decision Letter · Decision Letter 1]

28 Jul 2025

PGPH-D-25-00302R1

Trichiasis With and Without Tarsal Conjunctival Scarring: a multi-country observational study

Dear Dr. Habtamu,

Thank you for submitting your manuscript to PLOS Global Public Health. After careful consideration, we feel that it has merit but does not fully meet PLOS Global Public Health’s publication criteria as it currently stands. Therefore, we invite you to submit a revised version of the manuscript that addresses the points raised during the review process.

We look forward to receiving your revised manuscript.

Kind regards,

Javier H Eslava-Schmalbach, M.D., Ph.D., MSc

Academic Editor

Journal Requirements:

Additional Editor Comments (if provided):

Dear Authors,

This paper is relevant and highlights an avoidable condition that continues to be present in these countries.

We have received the results of the second review process for your manuscript. Thank you for incorporating the initial comments from the reviewers. As before, please provide an explicit response to each of the reviewers' comments: either accept, modify, or explain your position regarding each point.

Additionally, please consider my specific comments below to further improve your manuscript:

1. You state that this is a "multi-country comparative cross-sectional study." However, in the same section, you indicate the inclusion of cases and a group of non-cases matched by age, sex, and location. This approach is consistent with a matched case-control study. Please double-check the study design described in your methods section.

2. Based on the previous point, matching affects sample independence, resulting in dependent data. For this reason, standard ordinal logistic regression is not the most appropriate analytical approach for this design. I suggest you consider using a conditional ordinal logistic regression model or a generalized estimating equations (GEE) model suitable for dependent samples.

3. Although the sample is representative of the accessible population, cases were selected randomly from all available cases, and controls were matched to cases. At no point were individuals randomly sampled from the overall population. Therefore, I recommend avoiding the use of the term "prevalence" in the text; instead, use "frequency."

4. In the Methods, you indicate that "eyes" are the unit of analysis and that "eye-based analyses are adjusted for between-eye correlation." Please clarify exactly how this adjustment was performed statistically, and specify whether this adjustment accounts for the correlation between case and control groups.

5. There are some inconsistencies in the data presented in the tables that need to be addressed:

- In Tables 3 and 4, there are 220 comparison eyes in Ethiopia, but in Table 2, there are only 107 comparison subjects. Please review and correct this discrepancy.

- The total number of cases in Table 2 is 946. Assuming all cases have at least one trichiatic eye (946), and adding non-trichiatic eyes (Table 3: n=397), the total should be 1,343 eyes. However, in the column titled "Trichiasis case eyes" in Table 3, n=1,324. Please clarify these differences. As a reader, it is difficult to understand the source of these discrepancies. Please also double-check consistency across all other tables.

6. In Table 8, I recommend including the results (odds ratio and 95% confidence interval) for both the crude and adjusted analyses, so readers can see how the multivariable analysis alters the crude results.

Reviewers' comments:

Reviewer's Responses to Questions

**Comments to the Author**

1. If the authors have adequately addressed your comments raised in a previous round of review and you feel that this manuscript is now acceptable for publication, you may indicate that here to bypass the “Comments to the Author” section, enter your conflict of interest statement in the “Confidential to Editor” section, and submit your "Accept" recommendation.

Reviewer #1: All comments have been addressed

Reviewer #3: All comments have been addressed

2. Does this manuscript meet PLOS Global Public Health’s publication criteria ? Is the manuscript technically sound, and do the data support the conclusions? The manuscript must describe methodologically and ethically rigorous research with conclusions that are appropriately drawn based on the data presented.

Reviewer #1: Yes

Reviewer #3: Yes

3. Has the statistical analysis been performed appropriately and rigorously?

Reviewer #1: Yes

Reviewer #3: Yes

4. Have the authors made all data underlying the findings in their manuscript fully available (please refer to the Data Availability Statement at the start of the manuscript PDF file)?

Reviewer #1: Yes

Reviewer #3: Yes

5. Is the manuscript presented in an intelligible fashion and written in standard English?

Reviewer #1: Yes

Reviewer #3: Yes

6. Review Comments to the Author

Reviewer #1: General Assessment

This manuscript addresses a critical issue in global trachoma surveillance: the accuracy and consistency of field-based identification of trachomatous scarring (TS) in the diagnosis of trachomatous trichiasis (TT), and its implications for elimination thresholds. The research is based on a well-designed multi-country observational study conducted in Ethiopia, Uganda, and Nigeria. It directly informs debates around refining or revising the WHO’s TT definition in the context of elimination targets.

The topic is of high relevance to public health practitioners, epidemiologists, and policy-makers working on neglected tropical diseases (NTDs). The manuscript is generally well-written, the methodology is sound, and the findings are significant. However, some clarifications and expansions would strengthen the manuscript, particularly regarding methodological interpretation and policy implications.

Major Comments

Definition Clarity and Policy Implications

The paper should more clearly articulate the policy-level consequences of including or excluding TS from the TT definition. What would the implications be for countries approaching elimination? A short dedicated paragraph in the Discussion section outlining the public health trade-offs would improve the impact of the manuscript.

Grading Discrepancies and Training

The substantial discrepancy between field graders and expert assessments (e.g., Ethiopia NPV 7.2%) suggests systemic under-detection of TS. A more detailed analysis of the causes (e.g., grader training limitations, image-based vs. in-person evaluation differences) would be helpful. Was grader performance consistent across countries?

Justification of Reference Standard

Expert photographic grading is used as the gold standard. While appropriate, the manuscript should briefly justify its validity and potential limitations—especially where photographic grading might miss subtle clinical signs observable in person.

Statistical Analysis

The methodology around calculating the Negative Predictive Value (NPV) is clear, but the authors might consider presenting sensitivity and specificity as well for a fuller picture of misclassification. Even if not the focus, these metrics would support the robustness of findings.

Sample Size and Power

While the overall sample size is relatively robust, the number of “No TS” trichiatic eyes is relatively small in Uganda and Nigeria. The authors should comment on how this may affect the precision and generalizability of the NPV estimates from those sites.

Minor Comments

Abstract: The abstract cuts off abruptly and seems to be missing the final sentence(s). Ensure that it is completed, especially the results and conclusion components.

Acronyms: Consider defining TCS (tarsal conjunctival scarring) and TS (trachomatous scarring) clearly and consistently at first use to avoid confusion.

Figures/Tables: Ensure that raw numbers and percentages are clearly presented in tables, alongside confidence intervals for NPV. Consider including a summary table comparing field grader vs. expert grader discrepancies by country.

Grammar/Style: The manuscript is largely well-written, but some minor grammatical improvements could be made for fluency, particularly in the results narrative.

Recommendation: Minor Revision

This study makes a valuable contribution to the understanding of TT case classification and its implications for programmatic decision-making. With minor clarifications and expansion of interpretation, particularly regarding the operational and policy implications of the findings, the paper will be suitable for publication.

Reviewer #3: General Comments: Great efforts by the researchers to improve the manuscript by responding to all the queries and comments. The matching methodology was very robust in controlling for confounding variables related to demographic characteristics. The random selection has helped to reduce selection bias.

Specific questions related statistical analysis: 1) Were any sensitivity analyses conducted to test the robustness of the findings? If not, could they be included to strengthen the results? 2) How were missing data handled in the analysis, and what impact might this have on the study's conclusions?

4) Outcome measure: How was the reliability of the expert photographic grading assessed, and could inter-rater reliability be reported?

5) Discussion and interpretation: How do the findings compare with previous studies, and what are the implications for future research and public health interventions?

7. PLOS authors have the option to publish the peer review history of their article (what does this mean? ). If published, this will include your full peer review and any attached files.

**Do you want your identity to be public for this peer review?** For information about this choice, including consent withdrawal, please see our Privacy Policy .

Reviewer #1: **Yes: ** Dr. Ragni Kumari

Reviewer #3: No

---

## [Decision Letter · Decision Letter 2]

18 Sep 2025

Trichiasis With and Without Tarsal Conjunctival Scarring: a multi-country observational study

PGPH-D-25-00302R2

Dear Dr Habtamu,

We are pleased to inform you that your manuscript 'Trichiasis With and Without Tarsal Conjunctival Scarring: a multi-country observational study' has been provisionally accepted for publication in PLOS Global Public Health.

Best regards,

Julia Robinson

Executive Editor

Reviewer #1:

Reviewer #3:

Reviewer Comments (if any, and for reference):

Reviewer's Responses to Questions

**Comments to the Author**

1. If the authors have adequately addressed your comments raised in a previous round of review and you feel that this manuscript is now acceptable for publication, you may indicate that here to bypass the “Comments to the Author” section, enter your conflict of interest statement in the “Confidential to Editor” section, and submit your "Accept" recommendation.

Reviewer #1: All comments have been addressed

Reviewer #3: All comments have been addressed

2. Does this manuscript meet PLOS Global Public Health’s publication criteria ? Is the manuscript technically sound, and do the data support the conclusions? The manuscript must describe methodologically and ethically rigorous research with conclusions that are appropriately drawn based on the data presented.

Reviewer #1: Yes

Reviewer #3: Yes

3. Has the statistical analysis been performed appropriately and rigorously?

Reviewer #1: Yes

Reviewer #3: Yes

4. Have the authors made all data underlying the findings in their manuscript fully available (please refer to the Data Availability Statement at the start of the manuscript PDF file)?

Reviewer #1: Yes

Reviewer #3: Yes

5. Is the manuscript presented in an intelligible fashion and written in standard English?

Reviewer #1: Yes

Reviewer #3: Yes

6. Review Comments to the Author

Reviewer #1: Explanation of Review and Additional Comments for the Author

Thank you for the opportunity to review this well-conducted and timely manuscript, which explores a critical issue in the definition and operationalization of trachomatous trichiasis (TT) within global trachoma elimination programs.

The multi-country observational design encompassing Ethiopia, Uganda, and Nigeria is a major strength, allowing for broader generalizability across settings with high trachoma burden. The study is methodologically sound, with clearly defined outcomes and appropriate use of expert photographic grading to assess the presence of tarsal conjunctival scarring (TCS). The use of negative predictive value (NPV) as a key metric is appropriate and effectively demonstrates the discrepancy between field-based simplified grading and expert assessment.

The findings have important implications for the classification of TT in programmatic surveys. Specifically, the data suggest that relying on the presence of trachomatous scarring (TS) in field grading may lead to underestimation of TT burden, as many trichiatic eyes with no observed TS in the field were later found to have scarring upon expert photo grading. This insight is important for informing surveillance strategies and criteria for validating elimination of trachoma as a public health problem.

Importantly, the study also demonstrates that eyes with confirmed TCS tended to have more severe disease (entropion, trichiasis, conjunctival inflammation, and corneal opacity), reinforcing the clinical relevance of scarring in assessing disease severity and need for intervention.

Ethical and Publication Considerations

The study appears to have been conducted ethically, though confirmation of institutional and national ethics approvals should be clearly documented in the manuscript.

Informed consent procedures should also be briefly confirmed in the methods section for both trichiasis cases and comparison participants.

There are no apparent concerns regarding duplicate publication, data integrity, or conflicts of interest. If any portion of the data has been presented elsewhere (e.g., conference abstracts), this should be transparently disclosed.

Recommendation

Accept – No Revision Needed

The manuscript is scientifically rigorous, clearly written, and presents findings of direct relevance to policy and practice in trachoma elimination programs. No revisions are required.

Reviewer #3: Congratulations to the researchers.

7. PLOS authors have the option to publish the peer review history of their article (what does this mean? ). If published, this will include your full peer review and any attached files.

**Do you want your identity to be public for this peer review?** For information about this choice, including consent withdrawal, please see our Privacy Policy .

Reviewer #1: **Yes: ** Dr. Ragni Kumari

Reviewer #3: No
